# Acquired Triazole Resistance Alters Pathogenicity-Associated Features in *Candida auris* in an Isolate-Dependent Manner

**DOI:** 10.3390/jof9121148

**Published:** 2023-11-28

**Authors:** Flora Bohner, Csaba Papp, Tamas Takacs, Mónika Varga, András Szekeres, Joshua D. Nosanchuk, Csaba Vágvölgyi, Renáta Tóth, Attila Gacser

**Affiliations:** 1Department of Microbiology, University of Szeged, 6726 Szeged, Hungary; flora.bohner@gmail.com (F.B.); papp.cs66@gmail.com (C.P.); ttakacs.ttamas@gmail.com (T.T.); varga.j.monika@gmail.com (M.V.); andras.j.szekeres@gmail.com (A.S.); csaba@bio.u-szeged.hu (C.V.); 2Department of Medicine (Infectious Diseases), Albert Einstein College of Medicine, Bronx, New York, NY 10461, USA; josh.nosanchuk@einsteinmed.edu; 3Department of Microbiology and Immunology, Albert Einstein College of Medicine, Bronx, New York, NY 10461, USA; 4HCEMM-USZ Fungal Pathogens Research Group, Department of Microbiology, University of Szeged, 6726 Szeged, Hungary; 5HUN-REN-USZ Pathomechanisms of Fungal Infections Research Group, University of Szeged, 6726 Szeged, Hungary

**Keywords:** *C. auris*, antifungal resistance, triazoles, microevolution, virulence

## Abstract

Fluconazole resistance is commonly encountered in *Candida auris*, and the yeast frequently displays resistance to other standard drugs, which severely limits the number of effective therapeutic agents against this emerging pathogen. In this study, we aimed to investigate the effect of acquired azole resistance on the viability, stress response, and virulence of this species. Fluconazole-, posaconazole-, and voriconazole- resistant strains were generated from two susceptible *C. auris* clinical isolates (0381, 0387) and compared under various conditions. Several evolved strains became pan-azole-resistant, as well as echinocandin-cross-resistant. While being pan-azole-resistant, the 0381-derived posaconazole-evolved strain colonized brain tissue more efficiently than any other strain, suggesting that fitness cost is not necessarily a consequence of resistance development in *C. auris*. All 0387-derived evolved strains carried a loss of function mutation (R160S) in *BCY1*, an inhibitor of the PKA pathway. Sequencing data also revealed that posaconazole treatment can result in *ERG3* mutation in *C. auris*. Despite using the same mechanisms to generate the evolved strains, both genotype and phenotype analysis highlighted that the development of resistance was unique for each strain. Our data suggest that *C. auris* triazole resistance development is a highly complex process, initiated by several pleiotropic factors.

## 1. Introduction

In 2009, following the isolation of *Candida auris* from the external ear canal of an inpatient, Satoh et al. predicted this species to be pathogenic, based on its culturing characteristics and thermotolerance (as high as 42 °C) [1]. In the following years, *C. auris* has rapidly become one of the most relevant and notorious emerging fungal pathogens [2,3,4,5,6]. The fungus’ ability to develop antifungal resistance with high frequency, to survive on abiotic surfaces and colonize patients in the long term, and to be easily transmissible via direct human contact has enabled its rapid spread in healthcare settings [2,4,5,7,8,9,10,11]. Based on these factors, the US Center for Disease Control and Prevention (CDC) as well as the European Centre for Disease Prevention and Control (ECDC) refer to this pathogen as a serious global threat, and the World Health Organization (WHO) has classified *C. auris* as a critical priority pathogen [12,13,14]. To date, *C. auris* infections have been reported in more than 40 different countries from six continents [13,15].

Based on phylogenetic differences, *C. auris* was initially separated into four clades, with a fifth clade later reported in Iran [8]. These groups were originally associated with specific geographical locations: Clade I with South Asia, Clade II with East Asia, Clade III with Africa, Clade IV with South America, and Clade V with Iran [8,9,10]. Clades I, III, and IV were previously linked with invasive nosocomial fungal infection outbreaks, especially in the ICU setting. Generally, isolates from these clades are less susceptible to antifungals and have a remarkable potential to acquire multidrug resistance [11,16]. Fluconazole resistance is the most common (approx. 90%) among isolates, contributing to the global healthcare crisis *C. auris* has caused, especially in the developing world, where this antifungal is frequently the first empiric therapeutic choice for invasive fungal infection treatment [17,18]. The occurrence of multidrug resistance (triazole and polyene resistance) is quite prevalent (approx. 40%) [19]. PDR (pan-drug-resistant) isolates have also been identified, which is another signature of this species. Due to similar enzyme-binding abilities, cross-resistance development between fluconazole and voriconazole is a regular phenomenon in *Candida* species [6,17,20,21,22]. Among the commonly used triazoles, that leaves newer agents such as posaconazole and isavuconazole as alternative drugs available to treat systemic *C. auris* infections [23], although resistance to posaconazole has also been noted in clinical isolates [24].

Resistance mechanisms have recently been investigated, and processes similar to those of other *Candida* species have already been found in *C. auris* [25,26,27]. These mechanisms include (i) amino acid substitutions in drug target enzymes (e.g., Erg11) [6,28,29]; (ii) increased drug target production due to the corresponding genomic site’s duplication event (e.g., ERG11) [30,31]; (iii) point mutations in target associated genes (e.g., *ERG3*); or (iv) overexpression of ABC- and MFS transporter genes (e.g., *CDR1*, *SNQ2*, *MDR1*), thus increasing efflux pump activity [27,32,33]. Recently, Rybak et al. also verified distinct gain-of-function mutations in the zinc-cluster transcription factor *TAC1b*, which contributes to fluconazole resistance by elevating the expression of *CDR1* [34]. However, besides non-*auris*-specific resistance mechanisms, the presence of alternative drug-resistant processes was also suggested by previous studies [25,35].

In vitro microevolution is a widely accepted and frequently utilized method of studying diverse responses in microbes. In particular, it enables the monitoring of additional physiological alterations that might arise during drug resistance development, thereby allowing us to gain an insight into cost–benefit effects [36,37]. Similar experiments have previously been used for various *Candida* species to reveal unknown mechanisms of resistance [38,39,40,41]. These studies suggests that, in general, the development of triazole resistance attenuates virulence as an effect of a related fitness trade-off, although in some cases, the virulence of resistant strains is enhanced, further complicating the effective treatment of associated infections [42,43]. Microevolution experiments have already been successfully adopted in *C. auris* and they have been helpful tools to identify the genomic determinants of antifungal resistance [34,38,44,45]. Sequencing fluconazole-resistant microevolved 0381 and 0387 isolates provided evidence that after even a single passage with the antifungal, the acquisition of *TAC1b* mutations (F214S, R495G) can confer azole resistance [34,38]. A similar method was used on 17 clinical isolates to successfully identify azole resistance, conferring mutations in genes such as *IRA2* and *UPC2* [44]. This technique also revealed that antifungal pressure induces severe karyotype rearrangements in *C. auris* [45]. However, microevolution studies usually concentrate on fluconazole and reveal how short-term antifungal exposure effects fungal cells [38,39,46]. By using two additional triazoles (posaconazole and voriconazole) and a long-term in vitro experimental microevolution method, we aimed not only to uncover potentially *C. auris*-specific resistance mechanisms after protracted exposure to these different drugs, but also to reveal how acquired triazole resistance affects virulence attributes.

The purpose of this study was to investigate the effect of long-term triazole adaptation on two *C. auris* clinical isolates, originated from phylogenetically distant clades (East Asia and South Asia) [7]. The in vitro microevolution process that we previously developed for *C. parapsilosis* [40] was utilized to generate fluconazole-, posaconazole-, and voriconazole-evolved *C. auris* strains. Resistance-associated genomic changes were identified and the effects of the acquired resistance were investigated with a focus on the fitness and in vivo virulence attributes of each drug-evolved strain.

## 2. Materials and Methods

### 2.1. Strains and Culturing Conditions 

Prior to the experiments, strains were cultured in 5 mL YPD (1% D-glucose; 1% peptone; 0.5% yeast extract) liquid media supplemented with 1% penicillin–streptomycin, unless stated otherwise. After 24 h incubation with 200 rpm shaking at 30 °C, 150 μL aliquots of cells were inoculated into 5 mL fresh liquid media for another 16 h and incubated under similar conditions. Cells were then collected by centrifuging at 2400× *g* for 5 min and washed 3 times with 1× PBS. 

*C. auris* clinical isolates were maintained on solid YPD plates, while fluconazole (cat. no: PHR1160; Sigma-Aldrich, Burlington, MA, USA)-, posaconazole (cat. no: 32103; Sigma-Aldrich)-, and voriconazole (cat. no: 32483; Sigma-Aldrich)-evolved strains were cultivated in YPD agar plates supplemented with the respective antifungal drugs (64 μg/mL FLU; 16 μg/mL POS or VOR). Strains were stored at 4 °C when not in use. 

For the microevolution process, azole-sensitive *C. auris* strains were selected from the CDC and FDA Antibiotic Resistance Isolate Bank panel (https://wwwn.cdc.gov/arisolatebank/Panel/PanelDetail?ID=2, accessed on 29 November 2018). One isolate originated from the East Asian clade (Clade II), here referred to as 0381 (AR Bank # 0381, B11220 reference strain), while the other isolate belonged to the South Asian clade (Clade I) and is referred to as 0387 (AR Bank # 0387, B8441 reference strain).

### 2.2. Generation of Evolved Strains and Microevolution Process

The sensitivity of these strains to azoles was confirmed using the microdilution method performed according to the CLSI M27-A3 protocol [47]. 

The triazole (fluconazole [cat. no: PHR1160; Sigma-Aldrich], posaconazole [cat. no: 32103; Sigma-Aldrich], and voriconazole [cat. no: 32483; Sigma-Aldrich]) microevolution process was performed as described previously by Papp et al. [40] with minor modifications (Appendix A). Initial triazole concentrations were established as half of the MIC values determined by the CDC for the corresponding antifungal drugs in both susceptible isolates. Three individual cultures from both clinical isolates were adjusted to 0.1 at OD_600_ in liquid YPD media. Each culture was first incubated for 10 h (200 rpm shaking, 30 °C) in YPD medium without supplement; then, the initial concentrations of antifungals (Appendix A) were added, and cells were incubated for a further 14 h. Stock solutions for each antifungal were prepared as follows: 5 mg/mL fluconazole in water, 1 mg/mL posaconazole and voriconazole in DMSO. Stock solutions were further diluted in the early stages of the microevolution experiment to reach the final concentration. After the incubation period fungal cells were collected and re-cultured for 24 h three times in fresh media supplemented with the same concentration of antifungals. On the fourth day, the absorbance of the cultures was again adjusted to 0.1; then, with the same volume of antifungal drugs, cells were incubated for 10 h followed by 14 h incubation with a 2-fold concentration increase for each drug. After that, cells were collected and re-cultured in fresh media supplemented with the increased concentration of the antifungals. This two-fold increase in drug concentration for each strain was repeated every fourth day until the previously established final concentration (FLU: 512 μg/mL; POS and VOR: 32 μg/mL) was reached. A total of 50 μL of aliquots from each generated strain was plated to YPD agar media supplemented with the corresponding antifungals in the following concentrations: 64 μg/mL FLU; 16 μg/mL POS; 16 μg/mL VOR. Strains obtained directly from the microevolution experiment were only adapted to the presence of the corresponding drug, hence referred to as FLU^adp^, POS^adp^ and VOR^adp^. Aliquots from each adapted strain from the two initial isolates were then subcultured 10 times in fresh liquid YPD media for 24 h, without any drug supplement. These evolved strains are referred to as FLU^evo^, POS^evo^, and VOR^evo^. 

### 2.3. Antifungal Susceptibility Testing

Antifungal susceptibility tests were performed according to the M27-A3 protocol, with modifications regarding the range of tested concentrations and incubation temperature. MIC values were determined based on the M27-S4 supplementary material [47]. *C. auris* strains were defined as susceptible or resistant based on the tentative antifungal breakpoints previously suggested by the CDC [48], and extended by cutoff values for posaconazole and voriconazole proposed based on the dECOFF (epidemiological cutoff) derivalization method to determine MIC values higher than the wild-type distribution [49]. Inhibitory concentrations were determined using triazoles (fluconazole, posaconazole, voriconazole), echinocandins (anidulafungin (cat. no: SML2288; Sigma-Aldrich), caspofungin (cat. no: SML0425; Sigma-Aldrich), micafungin (cat. no: SML2268; Sigma-Aldrich)), and a polyene (amphotericin-B (cat. no: A4888; Sigma-Aldrich)). Antifungal susceptibility was measured in RPMI 1640 medium supplemented with L-Gln (cat. nr: 12-115F; Lonza) and MOPS (cat. nr: M3183; Sigma-Aldrich) after 24 h and 48 h. According to the standard protocol for echinocandins and azoles, MICs were defined at the lowest concentrations that result in a 50% decrease in growth capability, while MIC for amphotericin-B MIC was determined at 90% growth arrest. Incubation temperature (30 °C) was adjusted to recreate conditions used during the microevolution experiment. At least two individual experiments were performed, using three technical parallels for each. MIC values for each strain were listed in Table 1 and Table 2.

### 2.4. General Growth Kinetics

Synchronized fungal cultures were washed 3 times with 1× PBS. After washing, cell concentration was adjusted to 10^4^ cells/200 μL YPD in 96-well plates. OD was measured at 600 nm at both 30 °C and 37 °C. Growth curves were determined based on the mean OD_600 nm_ values of endpoint measurements at each time point. Three individual experiments were performed at both temperatures using eleven parallels for each strain. Analysis of the growth curves were performed in the R environment [50] using the ‘Growthcurver’ package [51] (Appendix A). 

### 2.5. Determining the Abiotic Stress Tolerance of the Evolved Strains by Spotting Assay 

For stress tolerance studies, overnight cultures of the initial and generated strains were collected and washed 3 times with 1× PBS. Cells were then serially diluted to 2 × 10^6^, 2 × 10^5^, 2 × 10^4^ and 2 × 10^3^ cells/mL concentrations; 5 μL of these dilutions was spotted into solid agar plates to reach final cell concentrations of 10^4^, 10^3^, 10^2^, and 10^1^. The growth of the generated evolved strains in the presence of osmotic stressors (glycerol, NaCl), cell-wall-perturbing agents (caffeine, calcofluor white, Congo red), and membrane detergent (SDS) was monitored and compared to the initial susceptible isolate. The following concentrations were used for each abiotic-stress-conferring agent: 8%, 10%, and 12% (wt/vol) glycerol; 1 M and 1.5 M NaCl; 12.5 mM, 15 mM, and 17.5 mM caffeine; 10 μg/mL, 25 μg/mL, and 50 μg/mL calcofluor white; 10 μg/mL, 25 μg/mL, 50 μg/mL, and 75 μg/mL Congo red; 0.02% (wt/vol), 0.04% (wt/vol), and 0.06% (wt/vol) SDS. The supplemented complex YPD agar plates were incubated at 30 °C and 37 °C for 48 h, and growth was compared to the initial clinical isolates that served as a control. Subjective scores were defined as described previously [52]. Experiments were repeated at least two times in each condition. Appendix A contains representative images of all plates complemented with stress-inducing agents in which growth capability changes were detected compared to the clinical isolate.

### 2.6. Cell Wall Staining

Cell wall staining was used to compare the cell wall composition of susceptible isolates and generated resistant strains. Cells were cultured, collected, and washed two times with 1× PBS. Pellets were then suspended in 200 μL 1× PBS + 1% BSA (cat. nr: A7906; Sigma-Aldrich) and incubated at 30 °C for 30 min with continuous rotation. Strains were washed 2 times with 1× PBS and suspended in 100 μL 1× PBS + 1% BSA containing cell wall dyes in the following concentrations: 8 μL 2.5 mg/mL ConA-FITC (concanavalin A A), 1 μL 1 mg/mL CW (calcofluor white), 1 μL 1 mg/mL WGA-TRITC (wheat germ agglutinin). Samples were incubated at 30 °C for 30 min with shaking. Stained cells were then washed 3 times with 1× PBS and suspended in 100 μL in 1× PBS, before being examined with a Zeiss Observer Z1 fluorescence microscope. For the Dectin-1, staining cells were incubated in 100 μL FCB (flow cytometry blocking) solution (0.5% BSA, 2 mM NaN_3_ in 1× PBS) for 30 min at 30 °C with shaking. Cells were then washed three times with 200 μL ice-cold FCW (flow cytometry washing) solution (0.5% BSA in 1× PBS) at low speed (0.8 g) for 2 min. Following the washing steps, samples were resuspended in 100 μL FCB solution supplemented with 1 μg/mL Fc (human): Dectin-1 (mouse) antibody and incubated for 60 min on ice. Cells were then washed with 200 μL of FCW three times at low speed, and pellets were resuspended in 100 μL of FCB solution containing Alexa Fluor 647-conjugated antihuman IgG Fc antibody in 1:200 ratio. Following 45 min incubation on ice, stained cells were washed with 200 μL FCW three times and resuspended in 50 μL of 1× PBS and kept on ice until use. For flow cytometry, the fluorescence intensity of 10^4^ cells was measured by Amnis FlowSight flow cytometer and analyzed by IDEAS 6.2 software. At least three individual experiments were performed for each strain. 

### 2.7. Efflux Pump Expression

RT-qPCR was used to monitor the relative expression of efflux pumps previously associated with azole resistance in *Candida* species. For total RNA isolation, overnight cultures of each strain were centrifuged and washed 3 times with 1× PBS. Cells were then collected and frozen in liquid nitrogen until use. RNA extraction was performed using the RiboPure RNA purification kit, yeast (cat. nr: AM-1926; Invitrogen, Waltham, MA, USA), according to the protocol provided by the manufacturer. To ensure the purity of the extracted RNA, additional DNase treatment with an RNase-free DNase set (cat. nr: 79254; Qiagen, Venlo, Netherlands) was applied according to the manufacturer’s recommendations. The absence of DNA contamination was checked with quantitative PCR (qPCR), prior to the use of the RevertAid first-strand cDNA synthesis kit (cat. nr: K1622; Thermo Scientific, Waltham, MA, USA). The Luna Universal qPCR Master Mix (cat. Nr: M3003L; NEB, Ipswich, MA, USA) quantitative PCR master mix in a CFX96 real-time system on a C1000 thermal cycler (Bio-Rad, Hercules, CA, USA) was used for the RT-qPCR experiments. *C. auris ACT1* was used as an internal control. Efflux pump coding genes were chosen based on the orthology of genes to other *Candida* species. RT-qPCR primers used in this study and the orthologs of the selected genes are listed in Appendix A. Three independent experiments were performed in triplicate in the case of each strain. Expression values were determined as the fold change, established by ΔΔCT threshold cycle (CT) analysis. Statistical analysis was performed by the Mann–Whitney test. Significant changes were noted if the *p*-value was <0.05.

### 2.8. Efflux Activity Assay

The detoxifying efficiency of the generated strains was measured by a Nile red reduction assay. Synchronized overnight cultures of the fungal cells were used and washed 3 times with 1× PBS. Cell concentration was adjusted to 10^8^/mL in 1× PBS. A total of 300 μL from each sample was treated with 50 μM enniatin and 10 mM of Na-azide to inhibit efflux processes. Samples were incubated for 1 h at 37 °C. After incubation, cells were washed 3 times with 1× PBS and stained with 10 μM Nile red. Strains were again incubated for 1 h at 37 °C and washed 3 times with 1× PBS. A total of 50 μL was taken from each sample for the 0 min measurement and kept on ice until flow cytometry (Amnis FlowSight) analysis. The remaining samples were suspended in 1× PBS containing 2% glucose to initiate the efflux processes. Every 30 min, 50 μL of cells was removed, washed twice (1× PBS), and kept on ice until measurement. The fluorescent intensity of 2 × 10^4^ cells was evaluated for each experiment. Three individual experiments were performed with each strain. Results were analyzed by IDEAS 6.2 software. Statistical analysis was performed by the Mann–Whitney test. Results were considered significant when *p*-value was <0.05.

### 2.9. XTT Reduction Assay

Alterations in biofilm forming abilities were studied by XTT assay. Synchronized fungal cultures were washed 3 times with 1× PBS. Cell concentration was set to 106 cells/mL in RPMI-1640 media and 100 μL of aliquots were added into 96-well microtiter plates using at least 9 parallels. An additional 100 μL of RPMI-1640 was added to the sample containing wells, and negative control wells also contained 200 μL of RPMI media. Depending on the experiment, microtiter plates were incubated for 24 or 48 h at 37 °C with 5% CO_2_. Following biofilm formation, plates were washed 2 times with 100 μL of 1× PBS to remove planktonic cells. A total of 100 μL of XTT solution (0.5 mg/mL dissolved in 1× PBS) (cat. nr: X6493; Thermo Scientific) supplemented with 10 mM menadione stock solution was added to the samples, with 2 wells left as a control for each strain. Microtiter plates were then covered to keep them from light and further incubated for 3 h at 37 °C with 5% CO_2_. Prior to reading absorbance, 80 μL of supernatant from each well was transferred into a new 96-well plate. Spectrophotometric measurement was performed at 490 nm. The absorbance of the control wells was subtracted from the XTT-containing wells. Three independent experiments were performed at 24 h and 48 h. Data analysis was performed by GraphPad Prism 7. Statistical significance was determined by the Mann–Whitney test. Results were considered significant when *p*-value was <0.05. 

### 2.10. Analysis of Sterol Composition by LC-HRMS

Fungal cells were cultivated overnight in YPD media at 30 °C with shaking and washed once with 1× PBS before the inoculation of 200 μL aliquots into 30 mL of YPD media and incubated for another night under the conditions mentioned above. Following incubation, cells were washed 2 times with 1× PBS and azole-challenged samples were treated for an additional 16 h with the following concentrations of antifungals: 256 μg/mL fluconazole; 16 μg/mL posaconazole; 16 μg/mL voriconazole. Cells were washed 2 times with 1× PBS, pelleted, and frozen. Further sample preparation was carried out as described by Varga et al. with certain modifications [53]. A total of 10 mg of freeze-dried samples was saponified with 2 mL of 10% KOH in methanol for 90 min at 80 °C. A total of 500 μL of water and 1 mL of *n*-hexane were added to cooled samples before being vortexed for 30 s. Following phase separation, the *n*-hexane phase was transferred into a 2 mL vial and dried under N_2_. *n*-hexane extraction was repeated twice. A total of 300 μL of methanol was used to dissolve the dried extracts before the samples were filtered through a polytetrafluoroethylene membrane filter with a pore size of 0.2 μm. Analyses of sterol compositions were performed in three biological parallels in the case of each strain. Appendix A contains raw data acquired during the analysis. 

### 2.11. Rapamycin Susceptibility Assay in Liquid Media

Synchronized fungal cultures were washed 3 times with 1× PBS. Cell concentrations were adjusted to 106 cells/mL in liquid YPD media. The highest concentration of rapamycin (1000 ng/mL) was prepared in YPD media using 10 mg/mL stock solution (diluted in sterile DMSO) (cat. nr.: 13346; Cayman Chemical Company, Ann Arbor, MI, USA). Starting with the adjusted concentration, twofold dilutions were serially prepared until reaching the lowest used concentration (1.95 ng/mL). A total of 100 μL of the prepared solutions was then dispensed into 96-well flat-bottom microdilution plates (from row 1 to 10). A total of 100 μL of the fungal cell solution was added to each well containing rapamycin, adjusting the final drug concentration to range from 500 ng/mL to 0.98 ng/mL. Row 11 wells were used as growth controls for each strain prepared from fungal cells and drug-free liquid YPD media supplemented with DMSO in an amount corresponding with the highest rapamycin concentration. Row 12 wells containing only the propagate media supplemented with rapamycin were used as a negative control. OD at 600 nm was measured by a plate reader at both 30 °C and 37 °C for 48 h. Endpoint data from 24 h, 36 h, and 48 h were analyzed further. Growth capacities were determined by normalizing the optical density values detected in rapamycin-treated cells to the corresponding drug-free control data for each experiment. In line with the literature [54], MIC values for rapamycin were defined as the drug concentrations where >80% growth arrest was detected compared to the corresponding drug-free control (Appendix A). Three individual experiments were performed at both studied temperatures (30 °C, 37 °C) using two parallels for each examined drug concentration. Data were evaluated using Graph Pad Prism 7 software. 

### 2.12. DNA Extraction for Whole-Genome Sequencing

For DNA extraction, fungal strains were incubated overnight in YPD media supplemented with 1% penicillin–streptomycin. Cells were pelleted by centrifugation (2.4 g; 5 min) and after discarding the supernatant, samples were dissolved in 500 μL lysis buffer. Glass beads ranging from 0.25 to 0.5 mm were used to disrupt fungal cells by vortexing for 3 min. A total of 270 μL ammonium acetate was added to the samples before incubation at 65 °C for 5 min. Samples were cooled down on ice for 5 min; then, 500 μL of chloroform–isoamyl alcohol (24:1) was added before samples were pelleted by centrifuging for 10 min at 16,200× *g*. The upper aqueous phase of each sample was carefully transferred into new sample tubes before isopropanol was added at a 1:1 ratio. Tubes were incubated at −20 °C for 45 min before centrifuging at 16,200× *g* for 10 min. A total of 500 μL of 70% EtOH was added to the samples before centrifuging for 5 min at 16,200× *g*. Pellets were dried at room temperature and dissolved in 25 μL of 1× TE. Samples were then stored at −20 °C until use. 

### 2.13. Genome Sequence Analysis

Whole-genome sequencing of the initial clinical isolates as well as of the generated triazole-evolved strains was performed by Novogene Co., Ltd. (Cambridge, UK) Strains were sequenced in three independent parallels. The quality and concentration of DNA samples were assessed by agarose gel electrophoresis and Qubit^®^ 3.0 fluorometer (Thermo Scientific) analysis. The construction of a DNA library was performed using the NEBNext^®^ DNA Library Prep Kit (NEB) following the instructions of the manufacturer. Quality control of the DNA libraries was performed by Qubit^®^ 2.0 fluorometer (Thermo Scientific) and Agilent^®^ 2100 bioanalyzer (Agilent, Santa Clara, CA, USA). Constructed DNA libraries were pair-end sequenced via Illumina^®^ platform, with the read length of PE150 bp at each end. 

Raw sequencing data were transformed to sequence reads by CASAVA software (1.8). The quality of the sequenced data was checked by Casava (1.8). Clean sequencing data were aligned with the reference sequences using BWA software (0.7.8-r455) and duplicates were removed by SAMTOOLS (0.1.19-44428cd). In the case of the 0381 clinical isolate and associated generated strains, the ASM301371v2 (GCA_003013715.2) genome was used as reference (mapping rate: 99.29%; average depth: 147.86–253.44×; coverage (1×): >99.62%; coverage (4×): 99.57%), while 0387 and associated strains were mapped to Cand_auris_B8441_v2 (GCA_002759435.2) (mapping rate: >98.48%; average depth: 156.59–259.51×; coverage (1×): >99.94%; coverage (4×): 99.92%). Individual SNP variations and InDels (≤50 bp) were detected using SAMtools (0.1.19-44428cd), structural variants (SV) (>50 bp) were detected by BreakDancer (1.4.4), and CNVnator (V0.3) was used to identify copy number variations (CNVs). Identified alterations were annotated by ANNOVAR (22 March 2015). Mutations that were present in all three sequenced parallels but were absent in their corresponding originated clinical isolates were further analyzed. Synonymous and nonsynonymous single-nucleotide polymorphisms (SNPs), nucleotide exchanges in noncoding regions, copy number variations, and structural variants were identified before the results were filtered. Nonsynonymous single-nucleotide variations (SNVs) and structural variations that were present in the generated triazole-evolved strains but not in the initial clinical isolates are listed in Table 3 and Table 4.

The functional effect of the detected amino acid substitutions was predicted using PROVEAN (Protein Variation Effect Analyzer) v1.1 software. Default score threshold (cutoff = −2.500) was used to determine deleterious variants.

### 2.14. In Vivo Murine Infection and Determining of Fungal Burden

To determine differences between the strains in terms of colonizing murine organs, 8–10-week-old female BALB/c mice were infected with fungal cells via the lateral tail vein. Overnight cultures of the fungal strains were washed 3 times, and cell concentrations were adjusted using 1× PBS. Each animal was infected with 100 μL of a fungal cell suspension containing 106 fungal cells. The mice were euthanized three days post infection, and their livers, spleens, kidneys, and brains were collected and homogenized. A total of 50 μL of each homogenate was plated into 1% penicillin–streptomycin-supplemented YPD agar plates. After 48 h incubation at 30 °C, colony-forming units (CFUs) were quantified on each plate. Colonization of the organs was calculated as the amount of CFUs determined in 1 g of the corresponding organ. Experiments were performed two times, with at least 4 animals infected with each strain per experiment (total of 10 mice/fungal strain). CFU data were analyzed using GraphPad Prism 7.00 software. Significant differences were determined by the Mann–Whitney test. Results were considered significant when the *p*-value was <0.05.

### 2.15. Statistical Tests

Statistical analysis was performed, and statistical significance was determined by GraphPad Prism 7.00 software. Unpaired t-tests and Mann–Whitney tests were used to evaluate the data. Results were considered as significant under the following *p*-values: *, *p* ≤ 0.05; **, *p* ≤ 0.005; ***, *p* ≤ 0.0005; ****, *p* ≤ 0.0001.

## 3. Results

### 3.1. Generation of C. auris Azole-Evolved Strains

Azole-evolved (evo) strains were generated as previously described by Papp et al. [40], with modifications. Briefly, both clinical isolates were incubated in drug-supplemented yeast extract–peptone–dextrose (YPD) liquid medium, containing increasing concentrations of each azole-type antifungal drug (Appendix A). To obtain evolved strains with stable, long-term antifungal-drug-adapted genomic alterations, strains were repeatedly cultured in YPD medium without any drugs. Thereafter, the generated fluconazole (FLU^evo^)-, posaconazole (POS^evo^)-, and voriconazole (VOR^evo^)-evolved strains were studied.

### 3.2. Antifungal Susceptibility of C. auris Azole-Evolved Strains

To test the antifungal susceptibility of the generated evolved strains, MIC values of azoles (FLU, POS, VOR), echinocandins (anidulafungin [AND], caspofungin [CAS], micafungin [MICA]), and a polyene (amphotericin B [AMB]) were determined (Table 1 and Table 2). Antifungal resistance was defined according to the tentative antifungal breakpoints for *C. auris* suggested by the CDC. Since cutoff values for posaconazole and voriconazole have not been determined yet, we used dECOFF cutoff values (1 µg/mL for voriconazole and 0.125 µg/mL for posaconazole) to distinguish MIC values that are above the wild-type distribution [49]. 

While all three 0387-evolved strains were nonsusceptible to the tested triazole drugs (FLU, POS, VOR) (Table 2), only the 0381 POS^evo^ strain showed substantially increased MIC values to both fluconazole (MIC > 512 μg/mL) and voriconazole (MIC > 32 μg/mL) (Table 1). According to the predicted dECOFF values, posaconazole and voriconazole tolerance of 0381 FLU^evo^ and VOR^evo^ were also above the wild-type distribution, but the clinical relevance of these data is unknown. It is important to note that all evolved strains of the 0387 isolate also showed cross-resistance to caspofungin, while anidulafungin MICs were also increased markedly (Table 2). We found a positive correlation between the increase in fluconazole and voriconazole MIC values in all of the strains. All *C. auris* azole-evolved strains remained highly susceptible to micafungin; only a twofold MIC value increase was detected in the 0387 FLU^evo^ strain after 48 h.

### 3.3. Growth Kinetics of the Evolved Strains in Complex Media

To study whether the microevolution experiment affected the growth capability of the resistant strains, we analyzed the growth curves of the generated strains at both 30 °C and 37 °C. The growth of the evolved strains was examined for 48 h in liquid complex media (YPD) and compared to the corresponding parental isolate (Figure 1 and Appendix A). According to the growth curve analyses, both the parental and evolved strains showed faster generation times (growth rates) and reached their first inflection point (maximum population growth, occurs at half of the carrying capacity) at earlier cycles when incubated at 37 °C (Appendix A). The preference for higher temperature was more prominent in the 0387 strains. The growth of 0381 azole-evolved strains was similar to that of the initial susceptible isolate (Appendix A). *C. auris* 0387 azole-evolved strains replicated faster (Figure 1) at earlier timepoints but reached the plateau phase at lower population densities than the clinical isolate at both 30 °C (Figure 1A) and 37 °C (Figure 1B). 

### 3.4. Triazole Resistance Development Alters Stress Tolerance in C. auris 

The abiotic stress tolerance of azole-evolved strains was examined on solid media under various conditions. Stressors were chosen to represent conditions that fungal cells encounter during host infection. These included osmotic and cell wall stressors and membrane-perturbing agents. The stress tolerance of the evolved strains was compared to that of their parental isolates, while differences in growth on YPD media were assessed by the evaluation of a spotting assay (Figure 2). 

Compared to its clinical parent isolate, the 0381 POS^evo^ strain was sensitive to the presence of caffeine; however, the FLU^evo^ and VOR^evo^ strains were tolerant to this agent at both temperatures (Figure 2A,C,D). All evolved strains were sensitive to high concentrations of calcofluor white (CW) compared to their parental strains. It is noteworthy that the POS^evo^ strain was also susceptible to a lower (20 μg/mL) CW concentration. Congo red (CR) sensitivity was also documented in this strain. Furthermore, this strain was unable to grow in the presence of sodium dodecyl sulfate (SDS)-supplemented media. These data suggest that in 0381 POS^evo^, both cell wall and membrane functions became impaired upon the acquisition of triazole resistance, since the susceptible clinical isolate tolerated these conditions better than the generated strain. 

All three 0387-evolved strains were less susceptible to the osmotic stressor NaCl at 30 °C than the clinical parent isolate (Figure 2B). Interestingly, these strains surpassed the growth of the parental strain in the presence of caffeine at both temperatures independent of the stressor’s concentration (Figure 2E,F). While the evolved strains showed growth defects on media containing CW at high concentrations (50 μg/mL), all tolerated the presence of CR (30 °C) better than the clinical isolate. Compared to the originating parental strain, the 0387 FLU^evo^ strain was more tolerant to SDS at 30 °C, whereas the POS^evo^ and VOR^evo^ strains were highly sensitive to this agent at both temperatures, suggesting an altered membrane composition (Figure 2E,F). Appendix A contains representative images of all plates complemented with stress-inducing agents in which growth capability changes were detected compared to the clinical isolate.

### 3.5. Azole Resistance Profoundly Alters the Ergosterol Content of C. auris Cell Membranes

Since triazoles inhibit sterol biosynthesis, we next examined the relative membrane sterol composition of the evolved and parental strains using liquid chromatography–mass spectrometry (LC-MS) (Figure 3) (Appendix A). 

Both FLU^evo^ strains showed a similar sterol composition to their corresponding parental strains, and ergosterol was the main component (>95% abundance). In both POS^evo^ strains, a profound depletion of ergosterol content was evident (approx. 80% and 99% decrease, for 0381 POS^evo^ and 0387 POS^evo^, respectively) (Figure 3A,B). An increased level of certain byproducts of the ergosterol biosynthetic pathway, such as fecosterol, episterol, ergostaenol, and ergostadienol, was detected (Figure 3G). In the 0381 VOR^evo^ strain, ergosterol remained the main sterol compound (98%) (Figure 3A), while in the 0387 VOR^evo^ strain, ergosterol (approx. 1%) was replaced by fecosterol/episterol/ergostadienol (approx. 90%), similarly to the POS^evo^ strains (Figure 3B). Interestingly, where ergosterol content decreased significantly, a previously unidentified sterol product (C_28_H_44_O_2_) was also detected. Based on carbon atom numbers and the prominent presence of ergostadienol and ergostaenol, the unknown component might be the result of a bypass pathway (Figure 3G). Taken together, acquired antifungal drug resistance can drastically alter the ergosterol biosynthetic pathway in *C. auris*, which appears to vary in an antifungal drug- or strain-dependent manner.

### 3.6. Upon Antifungal Challenge, the Activation of the Alternative Pathway Promotes the Appearance of a Yet Uncharacterized Sterol Product, C_29_H_46_O

The sterol composition of each parental and evolved strain was also examined in the presence of triazoles (Figure 3C,D). Strains were challenged with half of the drug concentration that had been reached during the microevolution experiment (half of the final drug concentration: 256 μg/mL fluconazole; 16 μg/mL posaconazole and voriconazole). As expected, in the drug-challenged parental strains, ergosterol depletion was high after fluconazole (approx. 15 and 27% in the 0381 and 0387 strains, respectively), posaconazole (approx. 18 and 26% in the 0381 and 0387 strains, respectively) and voriconazole treatment (approx. 18 and 26% in the 0381 and 0387 strains, respectively). In these strains, lanosterol and obtusifoliol were the most predominant sterol metabolites, with a prominent presence of 14-Me-fecosterol. The abundance of these byproducts indicates the activation of an alternative sterol biosynthetic pathway once *ERG11* is inhibited (Figure 3H). The appearance of the methylated metabolite, 14-Me-ergosta-dien-diol, as the terminal product of this alternative pathway indicates that the commonly applied triazoles promoted the accumulation of toxic sterol precursors.

In 0381 azole-evolved strains, drug treatment also correlated with a depletion of ergosterol content (approx. 40%, 1%, and 14% in the FLU^evo^, POS^evo^, and VOR^evo^ strains, respectively) (Figure 3C). In parallel, an increase in byproducts was noted. These included lanosterol/obtusifoliol (approx. 31%, 47% and 50%), 14-Me-fecosterol (approx. 9%, 42% and 14%) and the toxic byproduct, 14-Me-ergosta-dien-diol (approx. 5%, 1%, and 6% in FLU^evo^, POS^evo^, and VOR^evo^ strains, respectively). Based on this, it is evident that upon azole treatment, the depletion of ergosterol is compensated by the enrichment of lanosterol/obtusifoliol and 14- Me-fecosterol (especially in POS^evo^).

Following the azole treatment of 0387-evolved strains, ergosterol levels decreased severely (approx. 2% in FLU^evo^, absent in POS^evo^ and VOR^evo^) (Figure 3D). The reduction in ergosterol content was complemented by the buildup of lanosterol/obtusifoliol (approx. 55%, 40%, and 41%) and 14-Me-ergosterol (approx. 21%, 50%, and 48% in the FLU^evo^, POS^evo^, and VOR^evo^ strains respectively). Additionally, the purportedly toxic sterol byproduct 14-Me-ergosta-dien-diol was only present in the FLU^evo^ strain (approx. 8%), while being completely absent from both POS^evo^ and VOR^evo^ (Figure 3H,I).

Corresponding triazole treatment of *C. auris* 0381 strains and 0387 FLU^evo^ led to a mild build-up of a previously uncharacterized sterol product, C_29_H_46_O (Figure 3H). Based on the fact that this product was only detected in association with the presumably toxic compound 14-Me-ergosta-dien-diol, we suggest that their synthesis is closely related. Taken together, sterol composition data suggest that the adaptation process prominently differs between 0381- and 0387-originated strains.

### 3.7. Flow Cytometry Analysis Reveals Modifications in the Cell Wall of Evolved Strains

Altered stress responses in certain evolved strains indicated cell wall modifications. Therefore, flow cytometry (Figure 4) and fluorescent microscopy (Figure 5 and Appendix A) were used to further understand these changes in stress tolerance. 

Staining of the various cell wall components of 0381-derived strains revealed no significant alterations, although a slight, but not significant, increase in Dectin-1-Fc was detected, suggesting increased β-glucan levels in the FLU^evo^ and VOR^evo^ strains (Figure 4C). However, significant differences were detected in the 0387 group. For 0387 POS^evo^ and VOR^evo^, a higher number of cells had exposed chitin oligomers in their cell walls, based on the increased number of WGA-TRITC-positive cells in these populations, compared to their parental isolate (Figure 4B). The 0387 POS^evo^ and VOR^evo^ strains also showed significantly higher CW and WGA fluorescence levels, indicating an increased exposure of chitin and chitin oligomers compared to their parental isolate. The latter was also true for the 0387 FLU^evo^ strain, as we detected a significantly increased WGA fluorescent signal (Figure 4D). 

### 3.8. Sequence Analysis of the Evolved Strains Suggests That Multiple Distinct Resistance Mechanisms Can Develop during Antifungal Treatment

To identify genomic alterations potentially responsible for resistance development along with distinguishable phenotypic features, whole-genome sequencing was performed. Both the parental isolates and the generated strains were sequenced and compared to each other to find nonsynonymous variations and genomic modifications.

In the 0381 FLU^evo^ and VOR^evo^ strains, a single distinctive SNP event occurred at the coding region of transcription factor *TAC1b*. Additionally, the VOR^evo^ strain harbored a genome duplication at the genomic site including the coding region of Erg11. In the POS^evo^ strain, however, the only identified genomic variation was a deletion (200bp) in the coding region of B9J08_004089, an ortholog of the *C. albicans RFG1* and *S. cerevisiae ROX1* transcription factors (Table 3). *S. cerevisiae ROX1* is a mayor regulator of the hypoxic response, including ergosterol biosynthesis, and also has a role in pseudohypha formation, while *C. albicans RFG1* is solely responsible for filamentation and does not play a role in the regulation of hypoxia-related genes [55].

In the 0387-derived group, the same amino acid substitution occurred in the coding region ‘B9J08_002818′ in all three azole-evolved strains. According to the protein variation analysis, this mutation is predicted to be deleterious (score: −5.400). This gene is an ortholog of *BCY1* in both *C. albicans* and *S. cerevisiae* (Table 4). BCY1 is a key regulator of the fungal PKA pathway that is responsible for several crucial cell processes, such as ribosome biogenesis, adherence, biofilm formation, and morphogenesis, which can all potentially affect the antifungal resistance and virulence of *C. auris* [56,57,58]. In the FLU^evo^ strain, an additional presumably deleterious (score: −7.978) amino acid substitution (P95T) was identified in the coding region of B9J08_002259, an ortholog of *HGT4* responsible for the detection and transport of metabolites such as glucose and galactose in *Candida* species [59]. In the POS^evo^ strain, an SNP was also found in the *ERG3* (B9J08_003737) gene. The detected amino acid substitution (S316*) resulted in the insertion of a STOP codon (nonsense mutation), initiating the premature termination of Erg3 translation (score: −12.793).

### 3.9. Efflux Activity of the Triazole-Evolved Strains

Increased efflux pump activity of membrane transporters is one of the hallmarks of antifungal resistance in *Candida* species. Both ATP-binding cassette (ABC) transporters (Cdr1, Cdr2) and major facilitator superfamily (MFS) transporters (Mdr1) contribute to triazole resistance in *C. auris* [27,33]. To investigate the efflux-dependent detoxifying activity of evolved strains, we utilized a Nile red fluorescent probe. This dye is capable of penetrating fungal cells and accumulating under energy-deprived conditions. In *C. albicans*, Cdr1, Cdr2, and Mdr1 transporters are responsible for removing the probe from the cytosol [60]. To check the activity of these transporters in evolved *C. auris* strains, the staining of the cells was carried out under restrictive conditions (enniatin, Na-azide) and a high-concentration glucose pulse (100 mM) was applied to initiate the restoration of physiological efflux activity. The decrease in fluorescent signal was analyzed by flow cytometry.

The 0381 azole-evolved strains and their parental isolate showed comparable efflux activity values (Figure 6A). Interestingly, in 0387-originated strains, glucose treatment induced a rapid loss of median fluorescent signal (approx. 82%, 96%, and 97% for FLU^evo^, POS^evo^, and VOR^evo^, respectively), suggesting an almost immediate and complete removal of the dye (Figure 6B). These data suggest that alterations in the efflux activity of Cdr1, Cdr2, and Mdr1 might play a more substantial role in the antifungal resistance of 0387 strains. 

### 3.10. Expression of Efflux Pump Genes in Evolved Strains 

To determine which transporters might be responsible for elevated efflux activity in the evolved strains, we examined the relative expression of three ABC transporters (Cdr1, Snq2a, Snq2b) (Figure 7A,B) and two MFS-type efflux pumps (Mdr1, Tpo3) (Figure 7C,D). 

In the 0381 group, the expression of *CDR1* significantly increased in the FLU^evo^ and VOR^evo^ strains, but not in the POS^evo^ strain, which maintained expression levels similar to the parental isolate. Contrarily, the expression of both SNQ2 type pumps, *SNQ2a* and *SNQ2b*, significantly decreased in all three azole-evolved strains (Figure 7A). *MDR1* expression also decreased in both the FLU^evo^ and VOR^evo^ strains, but not in the POS^evo^ strain, where expression levels remained similar to the parental isolate. (Figure 7C). 

In the 0387 group, CDR1 expression was moderately higher in the 0387 FLU^evo^ strain, while a significant increase was seen in the 0387 POS^evo^ and VOR^evo^ strains. *SNQ2a* expression also increased in the POS^evo^ and VOR^evo^ strains. There was also a trend towards an increased SNQ2b expression in all three strains (Figure 7B). A remarkably high level of MFS transporter expression was detected in all azole-evolved strains compared to the parental isolate. *MDR1* expression increased approx. 30-, 70-, and 80-fold in the FLU^evo^, POS^evo^, and VOR^evo^ strains, respectively. *TPO3* expression levels also increased approx. twelvefold in the FLU^evo^ and fourfold in the VOR^evo^ strains (Figure 7D). 

### 3.11. Function Loss of the Protein Kinase Inhibitor BCY1 Plays a Role in Rapamycin Resistance of C. auris

Since decreased caffeine susceptibility (Figure 2B,E,F) and a deleterious amino acid substitution in the regulatory subunit of PKA (*BCY1*) (Table 4) occurred simultaneously in 0387 azole-evolved strains, it raised the possibility that the cAMP/PKA pathway could also be linked to TORC1 in *C. auris*, similar to observations in other fungal species, such as *S. cerevisiae* and *C. albicans* [61,62,63,64]. In these species, rapamycin, similar to the effects of caffeine, inhibits TORC1 (target of rapamycin complex 1). Thus, we examined the rapamycin tolerance of azole-evolved strains using multiple rapamycin concentrations, ranging from 0.98 ng/mL to 500 ng/mL (Figure 8). Compared to the parental isolate, the FLU^evo^ strain showed a clear resistance to rapamycin, whereas the POS^evo^ and VOR^evo^ strains were moderately tolerant to its presence at both 30 °C (Figure 8A) and 37 °C (Figure 8B). Appendix A contains the exact MIC values of rapamycin determined after 24 h, 36 h, and 48 h at 80% growth arrest. 

Based on these data, we hypothesize a constitutive activation of the PKA pathway might also influence the downstream functions of TORC1 through an unidentified intermediary.

### 3.12. XTT Assay Indicates an Association between Biofilm Formation and Acquired Azole Resistance

Based on the known downstream effectors of the cAMP/PKA pathway in *Candida* species, we hypothesized that *BCY1* mutations can affect the biofilm-forming ability of azole-evolved *C. auris* strains [57]. 

Based on the XTT assays performed at both 24 and 48 h, we found that for the 0381 group, the biofilm formation of the VOR^evo^ strain decreased significantly compared to thitse parental isolate. In contrast, after 24 h, the biofilm formation of both the FLU^evo^ and POS^evo^ strains increased significantly (Figure 9A), but this difference was not evident at 48 h (Figure 9B). 

For the 0387 group, all azole-evolved strains produced significantly higher amounts of biofilms compared to their parental isolate (Figure 9C,D). These results suggest that azole resistance development significantly altered the biofilm-forming ability of azole-evolved strains.

### 3.13. Acquired Triazole Resistance Significantly Alters the Virulence of Evolved Strains in Mice 

The virulence of evolved strains was examined in a mouse model of intravenous infection, wherein 8–10-week-old female BALB/c mice were challenged with the fungal strain via the lateral tail vein. After 3 days, the infected animals were euthanized, and the fungal burden of different organs was determined. 

Compared to the parental isolate, all 0381 resistant strains were present at significantly higher loads in the brain, while both FLU^evo^ and VOR^evo^ were less efficient at colonizing the spleen of the animals than the initial susceptible isolate (Figure 10A).

All 0387 azole-evolved strains showed decreased colony-forming abilities in the kidneys and the brains of the infected mice, suggesting a slight decrease in the virulence of these strains (Figure 10B). Although this was independent of the microevolution experiment, it is also important to note that the CFU values detected in the livers and spleens of the animals showed high variability. Based on these data, we hypothesize that under fitting circumstances, *C. auris* is capable to bypass the immune response in certain organs. 

## 4. Discussion

In vitro microevolution is an effective tool to study both the cause and the effect of antifungal resistance in *Candida* species. Often, antifungal evolution studies only focus on fluconazole, and they lack data about how acquired resistance affects the pathogenic potential of generated strains. As the increasingly common resistance of *C. auris* to fluconazole requires the utilization of alternative antifungals in clinical practice, such as posaconazole and voriconazole, our aim was to reveal possible mechanisms of resistance to various triazoles (FLU, POS, VOR) in two phylogenetically distant *C. auris* strains (0381 [B11220], 0387 [B8441]). Therefore, we challenged two azole-susceptible *C. auris* clinical isolates with increasing concentrations of drugs to generate resistant strains. These strains were then studied to reveal the possible consequences of antifungal adaptation. 

### 4.1. Experimental Evolution Can Induce Pan-Azole Resistance as Well as Caspofungin Cross-Resistance 

Based on antifungal susceptibility tests, all evolved strains gained stable resistance to the specific antifungal drug used during their microevolution process. Cross-resistance between fluconazole and voriconazole is common among *Candida* clinical isolates, and it is often associated with amino acid substitutions in *ERG* genes [65]. Consistent with this, we also demonstrated an increase in MIC values for fluconazole and voriconazole in all FLU^evo^ and VOR^evo^ strains. Posaconazole susceptibility differed between the two sets of strains. In the case of 0381 FLU^evo^ and VOR^evo^, posaconazole MIC was only slightly higher than the wild-type distribution, whereas 0387 FLU^evo^ and VOR^evo^ became more tolerant to this agent. 

In 2017, You et al. identified a *C. tropicalis* isolate that developed permanent pan-azole resistance following posaconazole prophylaxis due to increased expression of *MDR1* [66]. Based on this, we hypothesize that the excessively increased *MDR1* mRNA levels in the generated 0387 resistant strains might be the reason behind their pan-azole resistance. Besides resistance to all applied azoles, 0387 azole-evolved strains also acquired resistance to caspofungin, along with a moderate MIC increase for anidulafungin. The lack of *FKS1* mutation suggests the activation of alternative echinocandin resistance mechanisms. In *C. glabrata*, changes in membrane sphingolipid content were reported to modulate responses to both caspofungin and micafungin. Further studies regarding the lipid composition of the evolved strains are needed to determine the background of this unusual resistance phenotype. 

Notable changes were also detected in the sterol composition of the generated strains. Interestingly, the depletion of ergosterol in both 0381- and 0387-originated POS^evo^ strains and the 0387-derived VOR^evo^ strain did not confer amphotericin-B resistance. Previously, a similar effect was reported in *C. parapsilosis* [40]. The antifungal effectiveness of amphotericin-B in the absence of ergosterol has been previously described [67]. Depleted ergosterol content in these evolved strains (0381 POS^evo^, 0387 POS^evo^, 0387 VOR^evo^) also corresponded with their increased susceptibility to the detergent SDS due to disturbed membrane homeostasis. 

### 4.2. Spotting Assay Indicated Changes in the Cell Wall and Membrane Homeostasis of the Evolved Strains 

Compared to the clinical parental isolate, cell wall integrity in 0381 POS^evo^ was defective. Surprisingly, this sensitivity did not correspond with virulence experiments. The fact that the pathogenic potential of these strains did not substantially decrease suggests that this phenotype is specific to these experimental conditions and does not have significance during infection. Both 0381 FLU^evo^ and 0381 VOR^evo^ gained tolerance to the presence of caffeine. In *S. cerevisiae*, certain ABC-type transporters (Snq2, Pdr5) can utilize caffeine as a substrate, and, therefore, they can be involved in the detoxification of the cells [68]. Increased expression of *CDR1* (ortholog of *S. cerevisiae PDR5*) in the 0381 FLU^evo^ and 0381 VOR^evo^ strains suggests that this transporter could be responsible for the caffeine-tolerant phenotype. Similar observations were also reported in microevolved *C. parapsilosis* strains, where elevated efflux transporter expression was also suggested to be a potential reason for caffeine resistance [40].

### 4.3. Posaconazole and Voriconazole Evolution Induced Unique Sterol Composition 

When cultured in complex media, the 0381 FLU^evo^, 0381 VOR^evo^ and 0387 FLU^evo^ strains had sterol composition profiles comparable to both clinical isolates. However, the 0381 POS^evo^, 0387 POS^evo^ and 0387 VOR^evo^ strains showed a depletion or even a complete absence of ergosterol. These strains accumulated fecosterol/episterol/ergostadienol, indicating a partial restriction (0381 POS^evo^; 0387 VOR^evo^) or premature arrest (0387 POS^evo^) of the ergosterol biosynthetic pathway. In 0381 POS^evo^, this atypical sterol profile might be linked to the partial deletion we found in the B9J08_004089 gene. The *C. albicans* ortholog of this gene (*RFG1*) only regulates morphogenesis, but the same gene in *S. cerevisiae* (*ROX1*) is responsible for the repression of hypoxic genes, including *ERG* genes. Ollinger et al. confirmed that orthologs of *ROX1* can also play a role in the hypoxia response in *Candida* species, as the deletion of *ROX1* in *C. glabrata* was able to compensate for the loss of *UPC2* by regulating ergosterol-biosynthesis-related genes [69]. Because of the complex function of this gene, further experiments are needed to confirm that the detected partial gene deletion in 0381 POS^evo^ indeed contributes to antifungal resistance. Based on the molecular formula and the presumed dependence on ergostaenol, we also propose that the previously unclassified sterol product C_28_H_44_O_2_ (portensterol) could also be linked to this alternative pathway. 

In general, ergosterol is known to be necessary for the maintenance of membrane integrity. However, clinical isolates of *C. albicans* accumulating alternative byproducts have previously been found, showing that strains with atypical sterol profiles can be viable even in hostile environments [70,71,72].

### 4.4. Azole Treatment Promoted the Accumulation of a Yet Unclassified Sterol Product in C. auris

Following the antifungal challenge, the depletion of ergosterol content occurred in every strain where this component had previously been abundant. An accumulation of products synthesized upstream from the inhibited Erg11, such as lanosterol/obtusifoliol and 14-Me-fecosterol, was detected. Based on the ratio of 14-Me-fecosterol and its downstream toxic product, 14-Me-ergosta-dien-diol, the activation of the alternative pathway occurred in both the 0381 and 0387 clinical isolates. Upon antifungal treatment, 14-Me-ergostadien-diol was completely absent in 0387 POS^evo^ and 0387 VOR^evo^. Since only these two strains did not contain the unclassified component, C_29_H_46_O (stigmasterone), we suggest that this product might belong to this alternative pathway, probably upstream from the toxic 14-Me-ergostadien-diol. 

The toxic byproduct was absent in the 0387 POS^evo^ strain and, together with the absence of ergosterol, indicates a complete functional loss of C-5-desaturase (*ERG3*). In line with this, genome sequencing revealed a nonsense mutation (S316*) in the coding region of this enzyme. This modification results in a truncated product, and although 87% of the protein remains unaffected, based on the aforementioned reasons, the function loss of the enzyme is evident. Even though it represents a classical triazole resistance mechanism, only a few *C. auris* isolates with *ERG3* mutations have previously been identified. While the missense SNP, S58T, is not linked with antifungal resistance [73], nonsense mutation (W182*) can confer a reduction in susceptibility to azoles [38]. Burrack et al. recently identified another mutation, T227I, which led to high fluconazole tolerance in a *C. auris* clinical isolate [44]. In our case, the absence of the toxic sterol byproduct suggests that S316* indeed has a role in the triazole resistance of *C. auris*, although it is also important to mention that this strain also harbored an additional mutation in *BCY1.* Further studies in genetically modified strains are needed to prove the role of the identified SNP in triazole resistance. 

The sterol profiles of short-term fluconazole-evolved 0387 cells were studied in the past. This corresponded with our results regarding the 0381 FLU^evo^ strain (about 50% decrease of ergosterol content) but surprisingly differed from what we detected in 0387 FLU^evo^ (>90% ergosterol decrease) [34]. Although the generation of resistant strains essentially differed between the two studies, these data suggest that the development of resistance can be diverse, even within the same strain. 

### 4.5. Microevolution Significantly Changes Efflux Processes 

One of the most common mechanisms of azole resistance in *Candida* species is the increased expression of efflux pumps [74]. Genome analysis of *C. auris* recently revealed 20 ABC transporters, emphasizing the crucial role of these proteins. Expression analysis of these transporters indicated a consistently increased expression of *CDR1* and diffusely raised expression of *SNQ2*-type pumps in isolates with antifungal resistance [33]. A study by Rybak et al. further confirmed the relevance of *CDR1* as well as the possible role of *MDR1* in the resistance of *C. auris* to fluconazole [27]. In our generated strains, mRNA levels of *CDR1* were five- and sevenfold higher in 0381 FLU^evo^ and 0381 VOR^evo^, respectively. Genome sequencing analysis revealed mutations in the transcription factor-encoding gene, *TAC1B*, in both strains (N690S in FLU^evo^ and S19I in VOR^evo^). Previously, Rybak et al. also highlighted an association between the *TAC1B* mutation and an increased relative expression of *CDR1* in short-term fluconazole-adapted strains originated from the 0387 isolate. The role of *TAC1B* mutations in fluconazole and voriconazole resistance was confirmed by both gene deletion and by the introduction of the most commonly reported substitution (A640V) into azole-susceptible isolates [22,34]. An analysis of azole-resistant *C. auris* clinical isolates also supported this finding, as besides the *ERG11* mutation, *TAC1B* modification was also a prominently reported modification in *C. auris* [22]. Further experiments are needed to confirm the role of these amino acid substitutions (N690S, S19I) in the development of azole resistance.

Functional studies indicate that the increased expression of *CDR1* in 0381 FLU^evo^ and 0381 VOR^evo^ is not sufficient to extract Nile red from the cytosol. On the other hand, the extensively high expression of *MDR1* in the case of 0387-derived strains correlated with a rapid extraction of the fluorescent dye, indicating that Mdr1 can more effectively transport Nile red out of cells.

### 4.6. All 0387-Microevolved Strains Carry a Universal SNP in the PKA Pathway Inhibitor, BCY1 

Whole-genome sequencing of the evolved strains revealed a non-conventional mechanism that might also play a role in triazole resistance, as all three evolved strains originated from the 0387 isolate carried an amino acid substitution (R160S) in the *BCY1* (B9J08_002818) gene. *BCY1* is the regulatory subunit of the cAMP-dependent protein kinase (PKA) holoenzyme in both *S. cerevisiae* and *C. albicans*. Bcy1 can inhibit the PKA pathway by binding to the catalytic subunits Tpk1 and Tpk2 [57]. In *S. cerevisiae*, activation of the PKA pathway triggers several cellular processes involved in the regulation of cell growth and differentiation [57,61]. Besides these vital functions, the PKA pathway also regulates virulence-related downstream effectors in *C. albicans* and plays a role in white-to-opaque switch and morphogenesis. Several studies have linked the PKA pathway with the TORC1 (target of rapamycin complex 1) pathway [60,74]. Data by Martin et al. point out that the effects of TORC1 inhibition (rapamycin) can be suppressed by activating the Ras-cAMP PKA [75]. In 2015, Chatterjee et al., while analyzing the draft genome of *C. auris*, mentioned the potential importance of PKA, based on the possibility that activation of stress-signaling pathways can influence the tolerance of fungi towards fungicides and antifungal peptides [76]. Additionally, Zamith-Miranda et al. found the relative abundance of Tpk2 (target of Bcy1) to be higher in a fluconazole-resistant *C. auris* clinical isolate (MMC-1) compared to a susceptible strain (MMC-2) [77]. In *S. cerevisiae*, the deletion of *BCY1* was previously identified to also confer rapamycin resistance. Since rapamycin is a substance that, similar to caffeine, inhibits TOR signaling, it is highly probable that there is a yet uncovered connection between the PKA and TOR signaling pathways in *C. auris* [78]. Based on the cross-talk with TORC1, we hypothesize that extensive caffeine tolerance of 0387-derived evolved strains can be an effect of a LOF-inducing point mutation in *BCY1.* To prove this, we studied the rapamycin susceptibility of 0387-derived resistant strains. All three strains harboring a *BCY1* mutation tolerated the presence of rapamycin more than the clinical isolate. This phenotype was the most pronounced in 0387 FLU^evo^, while 0387 POS^evo^ and 0387 VOR^evo^ showed weaker rapamycin tolerance. This difference is probably a result of the alteration of the ergosterol biosynthesis pathway, since the lack of ergosterol is known to sensitize cells to a wide range of chemicals, such as rapamycin [79,80,81]. In our case, this might explain why the 0387 POS^evo^ and VOR^evo^ strains with irregular sterol profiles responded to rapamycin treatment more than FLU^evo^.

Another known consequence of the constitutive activation of Ras-cAMP signaling is that the affected cells fail to adopt to the environmental nutrient supply [63]. Our data support this observation as, according to growth kinetics results, the 0387 azole-evolved strains grew more rapidly at early timepoints in complex media and reached the plateau phase earlier and at lower cell concentrations than the clinical isolate, which suggests growth arrest due to nutrient depletion. 

In *C. albicans*, one of the most recognized downstream targets of the PKA pathway are the transcription factors Efg1 and Flo8. Both regulators are important in biofilm formation and they may have a role in both antifungal resistance development and virulence in *C. auris* [58,82]. In line with the literature, 0387-originated resistant strains harboring a *BCY1* mutation showed a significant increase in biofilm formation at both 24 h and 48 h. 

According to a recent study by Kim et al., the deletion of *BCY1* affects antifungal susceptibility by increasing the expression of *CDR1* and *MDR1*, perturbing ergosterol biosynthesis and increasing biofilm formation capability in *C. auris* [35]. Based on this, we can conclude that single-nucleotide polymorphisms (R160S) in *BCY1* can potentially appear as a result of a natural genome alteration upon long-term drug challenge and can be associated with a decreased susceptibility to antifungals. Additionally, in line with our data, constitutive activation of PKA pathway due to the deletion of *BCY1* has also been described to correlate with attenuated virulence in a systemic infection mouse model [35]. Nonetheless, genetic validation is still needed to confirm the role of the identified amino acid exchange.

### 4.7. Upon the Acquisition of Antifungal Resistance, C. auris Strains Can Increase Their Pathogenic Potential

To examine how acquired drug resistance might affect virulence, we also conducted in vivo experiments. Fungal burden in the brains of animals infected with 0381 azole-evolved strains was significantly higher compared to the parental isolate. Surprisingly, this increase was most prominent in the case of the brains of 0381 POS^evo^-infected mice, despite the severely impaired abiotic stress tolerance features and atypical sterol content of this strain. This might be also linked to the detected B9J08_004089 (*ROX1*) mutation, as oxygen concentration in the brain is slightly lower than in other organs; thus, uncontrolled expression of hypoxia-associated genes can provide a growth advantage in this tissue [83]. It is also important to note the significant CFU increase in the brain tissue of the animals, suggesting that upon antifungal resistance development, *C. auris* can acquire the ability to infect the central nervous system more effectively than susceptible isolates. 

Fungal burden in the kidneys and brains of mice infected with 0387 azole-evolved strains indicated an attenuated virulence compared to the parental isolate. Additionally, with VOR^evo^, we also detected a CFU decrease in the spleens, while a slight increase in fungal burden was registered in the livers. However, it is important to highlight that in the livers and spleens, the deviations of CFU values were high, suggesting that in certain circumstances, the infection can spread in these tissues in a supposedly uncontrolled manner. Although this hypothesis needs further experimental proof, it is strengthened by the observation that the early immune response in these two organs is highly similar upon *C. albicans* infection. According to Lionakis et al., significantly more neutrophils accumulate in spleen and liver tissues at the early stages of fungal infection than in other organs, and their numbers then decline over time. Therefore, if the pathogen somehow bypasses the initial short-term host response to infection, the disease may progress in an unrestricted manner in these organs [84]. The more effective clearance of the evolved cells in the kidney and brain might be due to significant changes in the exposure of cell wall components [85]. We hypothesize that higher exposure of chitin and chitin oligomers in 0387 azole-evolved strains could explain the more efficient immune cell activation. This observation, together with organ-specific attenuated virulence, also suggests the involvement of chitin in the recognition of *C. auris* cells [86]. 

## 5. Conclusions

Long-term microevolution in the presence of one of three commonly used triazole-type antifungals demonstrated the highly pleiotropic nature of antifungal development in *C. auris.* The acquisition of drug resistance in fungal species is generally associated with a fitness loss linked to a reduction in virulence. Our data on *C. auris* contrast with this dogma. Although we only worked with a limited number of strains, we showed that there is a possibility that severe virulence attenuation does not occur in *C. auris* upon acquisition of antifungal resistance. For example, 0381 POS^evo^, despite being pan-azole-resistant, colonized brain tissue to a greater extent than its clinical parental isolate. Furthermore, the SNP identified in the *ERG3* of the 0387 POS^evo^ strain highlighted that mutations in this gene can arise during posaconazole treatment, resulting in additional insights into the possible function of this gene in the azole resistance of *C. auris.* Our results also further support other works suggesting the importance of the identification and characterization of alternative resistance mechanisms, such as alterations in the PKA pathway. The presence of these nonclassical resistance mechanisms may provide an explanation for how *C. auris* can acquire resistance with such high frequency. Nevertheless, validation by molecular biology methods is required to ascertain the roles of these identified mutations in the development of resistance. However, it is also important to acknowledge that the microevolution process has some limitations. Our experiments with type strains may not be fully representative of the overall population and, as the evolved strains were generated in a controlled laboratory environment, it is uncertain how these data would transition to clinical settings. It is also important to highlight that characterizing a small number of strains can only indicate a limited number of possibilities for how antifungal resistance can develop, but does not give us definitive answers. Nevertheless, taken together, these findings provide novel insights into the complex nature of antifungal resistance development in this emerging species.

## Figures and Tables

**Figure 1 jof-09-01148-f001:**
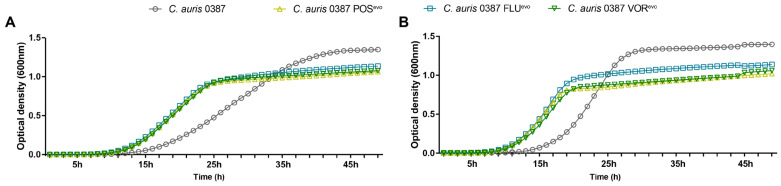
Growth kinetics of the generated strains compared to the parental isolates in complex media (YPD). (**A**) Growth curve of *C. auris* 0387 clinical isolate and evolved strains at 30 °C; (**B**) Growth curve of *C. auris* 0387 clinical isolate and evolved strains at 37 °C. Growth curves represent the means of at least 30 data points from three independent experiments.

**Figure 2 jof-09-01148-f002:**
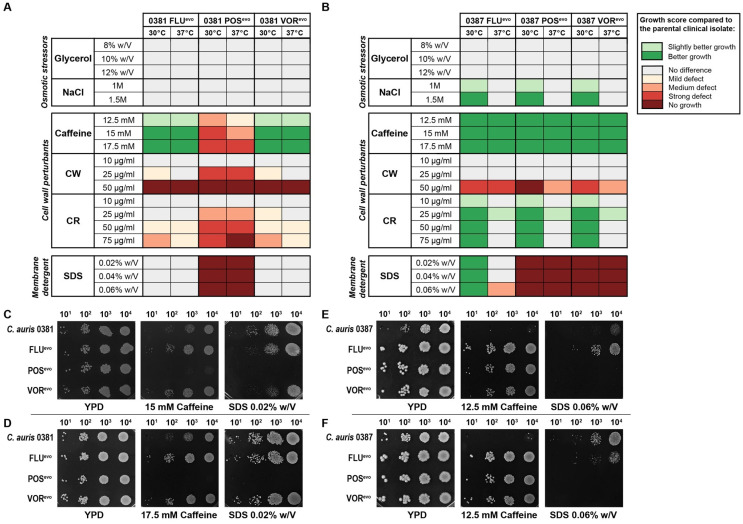
Abiotic stress tolerance of generated strains compared to clinical isolates. (**A**) Heat map representing stress responses of 0381-derived strains; (**B**) Heat map representing stress responses of 0387-derived strains. Prior to determining each growth score, differences in growth kinetics between the strains (based on YPD control plates) were also considered and the generated strains were then compared to the corresponding clinical isolate. Growth scores were determined from three independent experiments at both 30 °C and 37 °C; (**C**) Representative images of the growth of 0381-derived strains on YPD media, 15 mM caffeine, and 0.02% (*w*/*V*) SDS-supplemented media at 30 °C; (**D**) Representative images of the growth of 0381-derived strains on YPD media, 17.5 mM caffeine, and 0.02% (*w*/*V*) SDS-supplemented media at 37 °C; (**E**) Representative images of the growth of 0387-derived strains on YPD media, 12.5 mM caffeine, and 0.06% (*w*/*V*) SDS-supplemented media at 30 °C; (**F**) Representative images of the growth of 0387-derived strains on YPD media, 12.5 mM caffeine, and 0.06% (*w*/*V*) SDS-supplemented media at 37 °C.

**Figure 3 jof-09-01148-f003:**
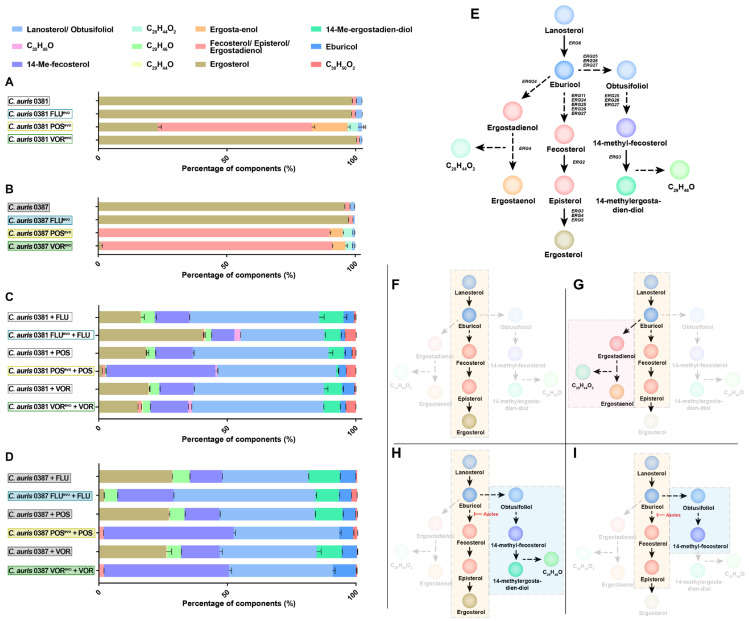
Membrane sterol composition of evolved strains with and without antifungal treatment. (**A**) Sterol compositions of 0381-derived strains and the clinical isolate were measured by LC-MS (liquid chromatography–mass spectrometry); (**B**) Sterol compositions of 0387-derived strains and the clinical isolate were measured by LC-MS; (**C**) Changes in membrane sterol composition of 0381-derived strains and the clinical isolate followed by antifungal treatment (half load of the MIC value reached as a result of the microevolution experiment [FLU: 256 µg/mL; POS: 16 µg/mL; VOR: 16 µg/mL]); (**D**) Changes in membrane sterol composition of 0387-derived strains and the clinical isolate followed by antifungal treatment (FLU: 256 ug/mL; POS: 16 ug/mL; VOR: 16 ug/mL); (**E**) Simplified schematic visualization of the predicted ergosterol biosynthetic pathway in *C. auris.* Color of circles corresponds with the color scheme used in sterol composition graphs; (**F**) Prediction of the ergosterol biosynthetic pathway in *C. auris* clinical isolates and both FLU^evo^ and VOR^evo^ strains. (**G**) Altered biosynthetic pathway marked with the depletion or complete absence of ergosterol was noted in the case of 0381 POS^evo^ and 0387 POS^evo^ and VOR^evo^ strains; (**H**) Following treatment with the utilized triazoles, clinical isolates (*C. auris* 0381 and 0387), 0381 FLU^evo^, 0381 POS^evo^, 0381 VOR^evo^, and 0387 FLU^evo^ showed a depletion of ergosterol content, accumulation of lanosterol/obtusifoliol, 14-Me-fecosterol, and the presence of toxic byproduct 14-Me-ergosta-dien-diol; (**I**) After treatment with posaconazole and voriconazole, respectively, 0387 POS^evo^ and 0387 VOR^evo^ showed complete absence of ergosterol accompanied by depletion of 14-Me-ergosta-dien-diol, suggesting premature arrest of alternative pathway.

**Figure 4 jof-09-01148-f004:**
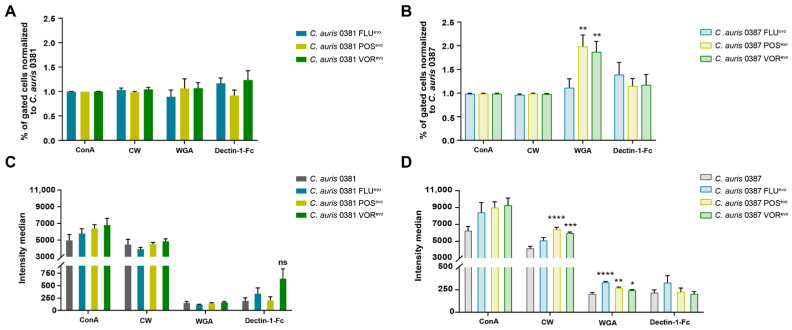
Cell wall alterations in generated strains. (**A**) Percentage of fluorescent positive cells in 0381-derived strains normalized to the susceptible isolate; (**B**) Percentage of fluorescent positive cells in 0387-derived strains normalized to the susceptible isolate; (**C**) Fluorescence intensity median of 0381 initial and generated strains; (**D**) Fluorescence intensity median of 0387 initial and generated strains. Fungal cells were stained by ConA-FITC (alpha-mannan), calcofluor white (chitin), WGA-TRITC (chitin oligomers), and Alexa 647 conjugated antihuman IgG Fc antibody (β-glucan) examined by flow cytometry. Statistical analysis was performed using one-way ANOVA with multiple comparisons. ConA: FITC-conjugated concavalin-A; CW: calcofluor white; WGA: TRITC-conjugated wheat germ agglutinin; Dectin-1-Fc: FC (human) Dectin-1 (mouse) (rec.) antibody linked with Alexa 647 conjugated antihuman IgG Fc secondary antibody; ns: non-significant. *, *p* ≤ 0.05; **, *p* ≤ 0.005; ***, *p* ≤ 0.0005; ****, *p* ≤ 0.0001.

**Figure 5 jof-09-01148-f005:**
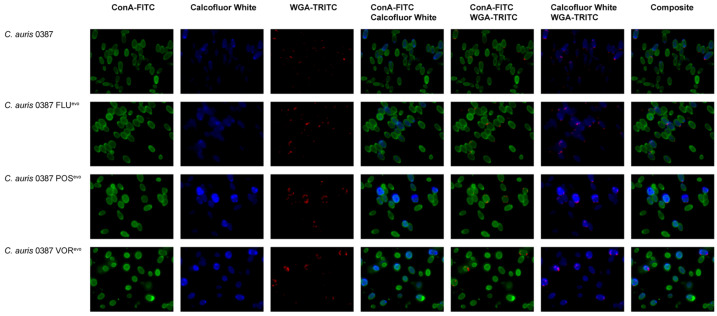
Fluorescence microscopic analysis of cell wall components of the 0387-isolate-derived strains. Alpha-mannan exposure of the cell wall was determined by staining with ConA-FITC; chitin was stained by calcofluor white; and chitin oligomer content was detected using WGA-TRITC. ConA: FITC-conjugated concavalin-A; WGA: TRITC-conjugated wheat germ agglutinin.

**Figure 6 jof-09-01148-f006:**
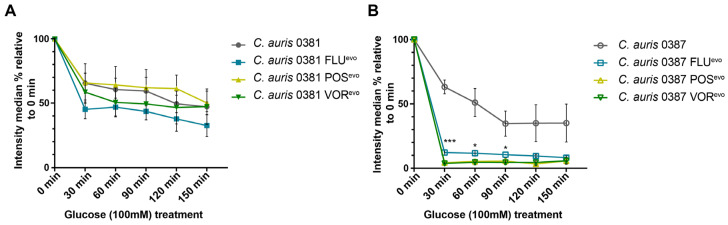
Efflux pump activity of generated resistant strains. (**A**) Detoxifying activity of efflux processes in 0381-derived strains examined by flow cytometry; (**B**) Detoxifying activity of efflux processes in 0387-derived strains examined by flow cytometry. Nile Red was used to label cells under energy-deprived conditions. After activation with glucose (100 mM), alterations in fluorescent intensity were monitored for 150 min. Median intensity percentages were calculated and compared to pre-pulse (0 min) median intensity of the corresponding strain. Statistical analysis was performed by unpaired *t*-tests at each examined time-point. *, *p* ≤ 0.05; ***, *p* ≤ 0.0005.

**Figure 7 jof-09-01148-f007:**
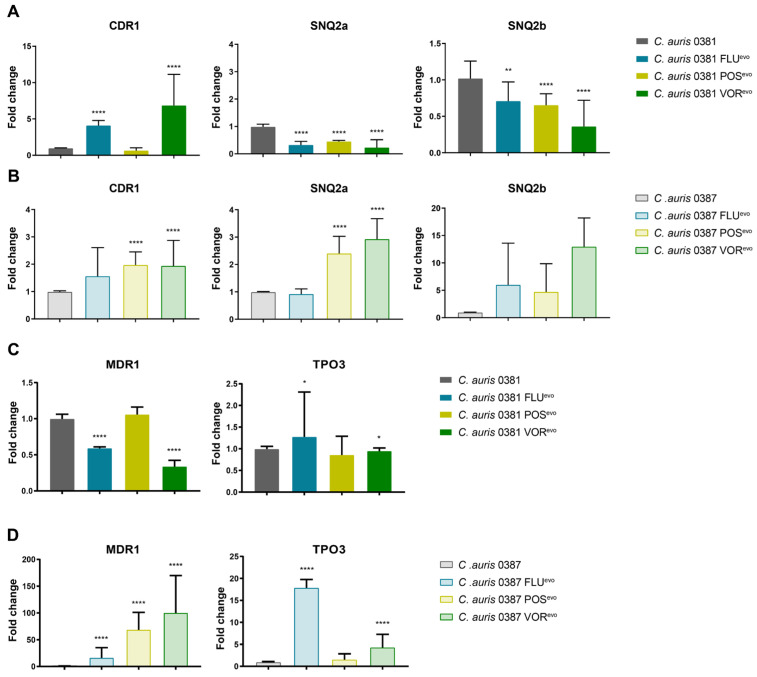
ABC and MFS efflux pump expression in evolved strains. (**A**) ABC-type transporter expression change in 0381-derived strains; (**B**) ABC-type transporter expression change in 0387-derived strains; (**C**) MFS-type transporter expression change in 0381-derived strains; (**D**) MFS-type transporter expression change in 0387-derived strains. *C. auris ACT1* was used as an internal control. Three independent experiments were performed each time in three technical parallels. Median values with interquartile ranges were used to plot the data. Statistical analysis was performed by the Mann–Whitney test. *, *p* ≤ 0.05; **, *p* ≤ 0.005; ****, *p* ≤ 0.0001.

**Figure 8 jof-09-01148-f008:**
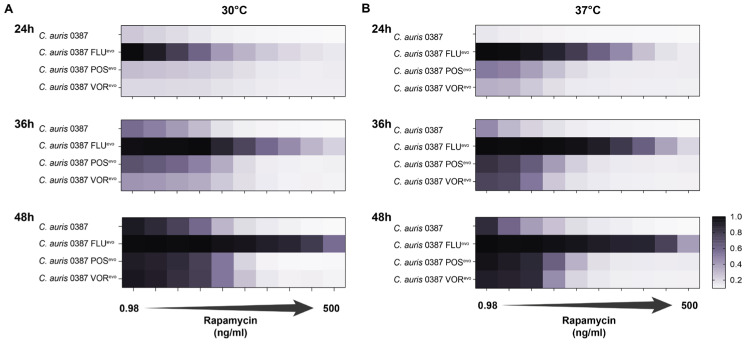
Rapamycin tolerance of 0387-isolate-derived triazole-resistant strains. (**A**) Rapamycin tolerance of the evolved strains at multiple timepoints (24 h, 36 h, 48 h) at 30 °C; (**B**) Rapamycin tolerance of the evolved strains at multiple timepoints (24 h, 36 h, 48 h) at 37 °C. A total of 500 ng/mL rapamycin was used as the maximum concentration and twofold dilutions were prepared until the minimum concentration (0.98 ng/mL) was reached.

**Figure 9 jof-09-01148-f009:**
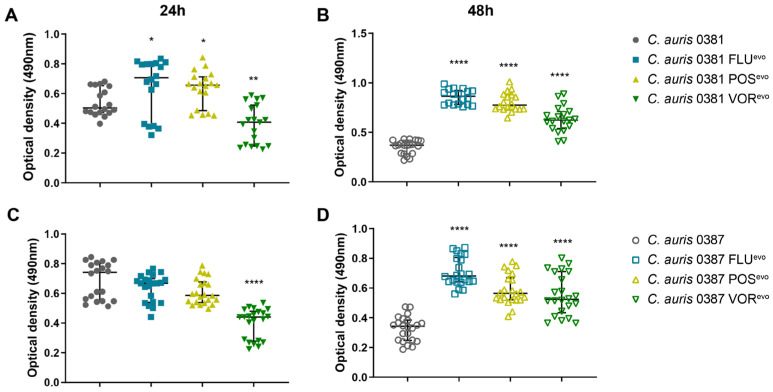
XTT reduction assay. (**A**) Biofilm-forming capability of the 0381 clinical isolate and evolved strains after 24 h incubation; (**B**) Biofilm-forming capability of the 0381 clinical isolate and evolved strains after 48 h incubation; (**C**) Biofilm-forming capability of the 0387 clinical isolate and evolved strains at 30 °C; (**D**) Biofilm-forming capability of the 0387 clinical isolate and evolved strains at 37 °C. At least 18 datapoints from three independent experiments were used in the case of each strain. Median values with interquartile ranges were used to plot the data. For statistical analysis, Mann–Whitney tests were used. *, *p* ≤ 0.05; **, *p* ≤ 0.005; ****, *p* ≤ 0.0001.

**Figure 10 jof-09-01148-f010:**
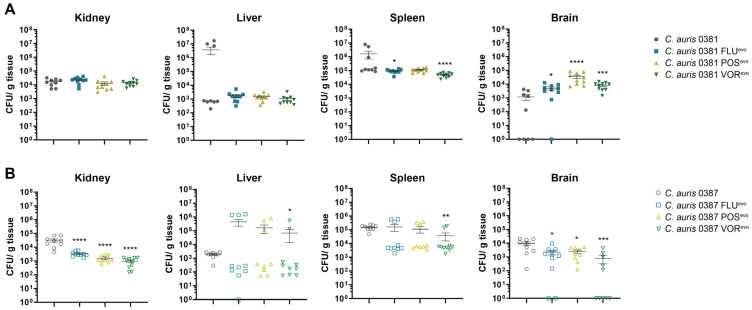
Virulence properties of azole-evolved strains. (**A**) Fungal colonization of kidneys, livers, spleens, and brains of animals challenged with 0381 clinical isolate and derived strains; (**B**) Fungal colonization of kidneys, livers, spleens, and brains of animals challenged with 0387 clinical isolate and derived strains. Two independent experiments were performed, using at least 4 animals each. Error bars indicate mean ± SEM. Significant differences were determined by Mann–Whitney tests. *, *p* ≤ 0.05; **, *p* ≤ 0.005; ***, *p* ≤ 0.0005; ****, *p* ≤ 0.0001.

**Table 1 jof-09-01148-t001:** Results of antifungal susceptibility testing following the microevolution process for the 0381-derived strains.

	MIC (μg/mL)													
Strains	AMB		FLU		POS		VOR		AND		CAS		MICA	
	24 h	48 h	24 h	48 h	24 h	48 h	24 h	48 h	24 h	48 h	24 h	48 h	24 h	48 h
*C. auris* 0381	0.25	1	8	16	0.0625	0.0625	0.0625	0.0625	0.03125	0.125	0.25	1	0.03125	0.03125
*C. auris* 0381 FLU^evo^	0.5	1	256 *	512 *	0.0625	0.25 **	2 **	4 **	0.0625	0.5	0.5	0.5	0.03125	0.03125
*C. auris* 0381 POS^evo^	0.5	0.5	>512 *	>512 *	>32 **	>32 **	>32 **	>32 **	0.125	0.5	0.25	0.25	0.03125	0.03125
*C. auris* 0381 VOR^evo^	0.5	1	>512 *	>512 *	0.25 **	0.5 **	8 **	16 **	0.0625	0.125	0.25	0.25	0.03125	0.03125

* Plausible emergence of antifungal resistance, based on the breakpoints suggested by the CDC. ** Plausible resistance to posaconazole and voriconazole based on dECOFF derivatization method.

**Table 2 jof-09-01148-t002:** Results of antifungal susceptibility testing following the microevolution process for the 0387-derived strains.

	MIC (μg/mL)													
Strains	AMB		FLU		POS		VOR		AND		CAS		MICA	
	24 h	48 h	24 h	48 h	24 h	48 h	24 h	48 h	24 h	48 h	24 h	48 h	24 h	48 h
*C. auris* 0387	0.25	0.5	2	8	0.0625	0.0625	0.0625	0.0625	0.0625	0.125	0.5	1	0.03125	0.03125
*C. auris* 0387 FLU^evo^	0.25	0.5	>512 *	>512 *	>32 **	>32 **	16 **	>32 **	0.5	1	>16 *	>16 *	0.03125	0.0625
*C. auris* 0387 POS^evo^	0.5	0.5	>512 *	>512 *	>32 **	>32 **	>32 **	>32 **	0.125	0.5	2 *	4 *	0.03125	0.03125
*C. auris* 0387 VOR^evo^	0.5	0.5	>512 *	>512 *	>32 **	>32 **	>32 **	>32 **	0.25	0.5	2 *	2 *	0.03125	0.03125

* Plausible emergence of antifungal resistance, based on the breakpoints suggested by the CDC. ** Plausible resistance to posaconazole and voriconazole based on dECOFF derivatization method.

**Table 3 jof-09-01148-t003:** Identified modifications in *C. auris* 0381 azole-evolved strains.

***C. auris*** **0381 FLU^evo^**					
Type of mutation	Gene	*C. albicans* ortholog	*S. cerevisiae* ortholog	Amino Acid Substitution	Function
Nonsynonymous SNP	B9J08_004820 (*TAC1b*)	*HAL9*	*HAL9*	N690S	Putative transcriptional activator of the drug-responsive genes CDR1 and CDR2; gain-of-function mutations are associated with resistance to fluconazole
***C. auris*** **0381 POS^evo^**					
Type of mutation	Gene	*C. albicans* ortholog	*S. cerevisiae* ortholog	Amino Acid Substitution	Function
Deletion (200bp)	B9J08_004089	*RFG1*	*ROX1*	-	Ortholog(s) has DNA binding, bending, DNA-binding transcription repressor activity, RNA polymerase II-specific, cis-regulatory region sequence-specific DNA binding, sequence-specific DNA binding, transcription corepressor activity
***C. auris*** **0381 VOR^evo^**					
Type of mutation	Gene	*C. albicans* ortholog	*S. cerevisiae* ortholog	Amino Acid Substitution	Function
Nonsynonymous SNP	B9J08_004820 (*TAC1b*)	*HAL9*	*HAL9*	S19I	Putative transcriptional activator of the drug-responsive genes CDR1 and CDR2; gain-of-function mutations are associated with resistance to fluconazole
Duplication	CJI96_0001194-CJI96_0001248 *			-	

Asterisk indicates the genome section containing *ERG11*.

**Table 4 jof-09-01148-t004:** Identified modifications in 0387 azole-evolved *C. auris*.

***C. auris*** **0387 FLU^evo^**					
Type of mutation	Gene	*C. albicans* ortholog	*S. cerevisiae* ortholog	Amino Acid Substitution	Function
Nonsynonymous SNP	B9J08_002818	*BCY1*	*BCY1*	R160S	Ortholog(s) have cAMP binding, cAMP-dependent protein kinase inhibitor activity, cAMP-dependent protein kinase regulator activity
Nonsynonymous SNP	B9J08_002259	*HGT4*	*SNF3*	P95T	Ortholog(s) have role in detection of glucose, fructose transmembrane transport, glucose-mediated signaling pathway, glucose transmembrane transport, mannose transmembrane transport, negative regulation of meiotic nuclear division
***C. auris*** **0387 POS^evo^**					
Type of mutation	Gene	*C. albicans* ortholog	*S. cerevisiae* ortholog	Amino Acid Substitution	Function
Nonsynonymous SNP	B9J08_002818	*BCY1*	*BCY1*	R160S	Ortholog(s) have cAMP binding, cAMP-dependent protein kinase inhibitor activity, cAMP-dependent protein kinase regulator activity
Nonsynonymous SNP	B9J08_003737	*ERG3*	*ERG3*	S316*	Ortholog(s) have C-5 sterol desaturase activity, role in ergosterol biosynthetic process and endoplasmic reticulum lumen localization
***C. auris*** **0387 VOR^evo^**					
Type of mutation	Gene	*C. albicans* ortholog	*S. cerevisiae* ortholog	Amino Acid Substitution	Function
Nonsynonymous SNP	B9J08_002818	*BCY1*	*BCY1*	R160S	Ortholog(s) have cAMP binding, cAMP-dependent protein kinase inhibitor activity, cAMP-dependent protein kinase regulator activity

## Data Availability

SRA data for all of the generated strains can be accessed under the following reference number: PRJNA843706. Biosample references for 0381-originated generated strains can be accessed using the reference numbers SAMN28766907 for FLU^evo^, SAMN28766908 for POS^evo^, and SAMN28766909 for VOR^evo^. Biosample references for 0387-derived generated strains can be accessed using the reference numbers SAMN28766910 for FLU^evo^, SAMN28766911 for POS^evo^, and SAMN28766912 for VOR^evo^. Sequence data of the clinical isolates can be accessed under the following reference numbers: SAMN37453806 (B11220, 0381), SAMN37453807 (B8441, 0387).

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
