# Peer review of "Acquired Triazole Resistance Alters Pathogenicity-Associated Features in *Candida auris* in an Isolate-Dependent Manner"

_jof, 2023, doi:10.3390/jof9121148_

Round 1

Reviewer 1 Report

Comments and Suggestions for Authors

The work of Bohner et al. aims at disclosing the complex and intertwined mechanisms governing acquired resistance to azoles in the emerging pathogen C. auris. In my view, the results presented by the authors confirm this complex nature since it is likely that this process is pleiotropic and involves many players. The research methodology was appropriate in this work, however, by trying to link all things together in my opinion the authors end up inducing quite a significant amount of speculation and the MS misses a message that should stay in the end on what were the real and novel findings of this work. The phenotypic profilings undertaken (for stress, for example or biofilms) use exprimental settings that are different from those used in the evolution experimnts. This will influence the response of the cells and ncessarily the way of this environmental influnce is exerted will depend on the genetic background of the strains which is different in the evolved vs non-evolved strains. THis is something that it is very difficult to establish since there will be interactions which we can't predict. How do you know if the results obtained for the strains result from the modifications they had acquired along azole-resistance evolution or from an influence exerted by the environment that is now different in these also different strains (meaning it can result from indirect effects). The way the message is provided turns the MS hugely complex and a bit too much speculative. 

1) It is difficult to understand the strain nomenclature in the abstract because we have not yet read the MS. Thus, likely the abstract should introduce readers to these unclear names. It is very unclear to which uncommon (uncommon in the species? in the strain? in th genus?) resistance mechanisms the authors refer to that were disclosed in this work. In my opinion the overall abstract should be reformulated and convey readers in a more accurate manner what were the results avoiding broad and generalistic descriptions that might prevent comprehension.

2) It seems that the authors have used strains that have been used before by others. Is this the case? If yes clarify on how your results differ from those. And how do your results differ from other C. auris strains that had been evolved for resistance?

3) Strain 0387 appears to have an already growth deffect while growing in unsupplemented YPD. Can't this influence the quantification of MIC since we base this on the decrease in growth, comparing precisely with the control? In other words strains get resistant in an easy manner because if they grow poorly they can surpass the treshold easily. This phenomenon has been observed in C. albicans resulting in sub-sets of populations and antifungal tolerance. Can't this be impactful here as well?

4) It is not clear what was the objective of the authors with the experiments shown in FIg.1. What was the rationale for this? Wouldn't this be better in supplementary material? By the way, growth curves should be log and the differences between the strains are very slight (with the exception of the growth deffect of strain 0387) which is more visible.

5) Cell wall perturbations and membrane perturbations were performed using an experimental setting different from the one used to evolve the strains. This may influence the response of the cells in a manner that is unrelated with the azole-resistance phenotype

6) In line 508 authors aim to make the point that the abundance of byproducts suggests there is an alternative pathway to Erg11 when the cells are challenged with azoles. But...don't we know that this is the case and the reason why cells accumulate the methylated sterols? What is the point that the authors are trying to make here? This figure (which has very intersting results) is still very complex, maybe if the authors present the ratio btween what happns to sterols in the same strain after exposure to FLC and before this would become easier to follow. The identification of the new sterol seems very intersting, has this been described in other species?

7) while in the abstract argues about the multiplicity of mechanisms governing acquisition of azole resistance, the title of the section describing the genomics section argues the opposite. In fact, since the genes affected are not the same why the authors make this point for a single mechanisms of resistance? 

8) Informatic predictions on how modifications affect proteins are much flawed and, in my view, should be disregarded specially in transcription factors (and the authors make this point in the discussion). Have these modifications in Tac1B been described elsewhere? in its homologues in other yeasts? To clearly associate these modifications with changes in protein activity, it is essential to undertake the necessary complementary functional assays using the wt Tac1B and the mutated one. Without that, it appears only as a speculation to say that it might be affected. What are the evidences for that?

9) A deletion of 200 bp in 004089 gene was inside the coding sequence? If yes, this surely has an impact in protein activity! THis is wrongly presentd and must be explained so that we understand what happens to this protein, is it inactive? or active? Specially important considering the role of Rox in ergosterol metabolism in other yeasts. 

10)What it means an X substitution? Is the protein truncated? if yes, clarify this aspect please. what percentage of the protein remains ?

11) BCY1 is not a transcription factor...

12) The discussion section is so long...it is very very difficult to understand what the authors want to convey as it seems that all things have to be connected. Please try to reduce this and aim for a message that we can take out this very interesting set of results. 

Comments on the Quality of English Language

The paper needs to be revised for comprehension and there are citations missing in some parts (discussion mainly).

Author Response

We thank the reviewers and the editor for their time and effort to evaluate our manuscript carefully. We appreciate all the comments and concerns raised during the review and hope our answers effectively address each point. 

1) It is difficult to understand the strain nomenclature in the abstract because we have not yet read the MS. Thus, likely the abstract should introduce readers to these unclear names. It is very unclear to which uncommon (uncommon in the species? in the strain? in th genus?) resistance mechanisms the authors refer to that were disclosed in this work. In my opinion the overall abstract should be reformulated and convey readers in a more accurate manner what were the results avoiding broad and generalistic descriptions that might prevent comprehension.

Thank you for the valuable insight, we have now reformulated the abstract as requested.

2) It seems that the authors have used strains that have been used before by others. Is this the case? If yes clarify on how your results differ from those. And how do your results differ from other C. auris strains that had been evolved for resistance?

We thank the reviewer for the comment. For reproducibility, we are using type strains. To clarity, the 0381 isolate is the first identified C. auris isolate while 0387 is another frequently used isolate for fundamental investigations. Both are commonly applied by several groups working with this species. The reason why we also stick to their use was simply due to the fact that they were the only azole susceptible strains in our strain collection with at least a partly annotated genome at the time of the project’s beginning. For microevolution experiments it is essential to seek drug sensitive strains in order to successfully generate the microevolved strains we aimed to comprehensively study and thus reveal yet unrevealed antifungal resistance mechanisms in this species. Different groups use different protocols to obtain resistant strains with differing results. The more common features and mechanisms we find the more we are certain of the presence of a conserved resistance mechanism. On the other hand, these studies could also supplement each other, by revealing uncommon mechanisms, or mechanisms that are specific to a certain condition, or only to a group of isolates e.g. isolates derived from a specific body site. Ultimately, all these studies contribute to the bigger picture that we seek to understand. As it is known that almost every disease is multifactorial, changes in microbial physiology could also be multifactorial. With such studies, we seek to identify these factors. Only this way can we have a change to fully understand resistance mechanisms, and to come up with effective solutions.

Our C. auris microevolution method was based on our previous work we performed in C. parapsilosis. Although microevolution experiments usually have similarities, to our knowledge our method differs from others in two major points. First, in contrast with short-term experiments we keep the antifungal pressure on for a long time, and by gradually increasing the concentration of the drug. Second, following the adaptation process we stabilize the resistant phenotype by serial inoculation without antifungal pressure, to rule out epigenetic changes due to short term adaptation. To be more straightforward we provided a detailed explanation in the aims section. As microevolution protocols vastly differ their direct comparison is to be handled carefully. We now supplemented the discussion part of the MS to address this question.  

3) Strain 0387 appears to have an already growth deffect while growing in unsupplemented YPD. Can't this influence the quantification of MIC since we base this on the decrease in growth, comparifng precisely with the control? In other words strains get resistant in an easy manner because if they grow poorly they can surpass the treshold easily. This phenomenon has been observed in C. albicans resulting in sub-sets of populations and antifungal tolerance. Can't this be impactful here as well?

Different C. auris clinical isolates have different characteristics, thus may differ in their growth abilities. So is the case for 0387 and 0381, both of which effectively grow in complex YPD medium, however one (0387) has a slightly lower replicating rate, than the other (0381). This is a natural phenomenon, given the distinct nature of the isolates. When determining the MIC values of the evolved strains, we compared the inhibitory cc of each drug - that was required to eliminate a certain ratio of cells (50% for azoles and echinocandins, 90% for amphotericin B) of the population - of these strains to that of the parental (or originating) strain, to see if the drug microevolution was successful, and if so, to what extent? We further aimed to see if resistance development to a drug substance resulted in cross-resistance to another. In reality, these MICs were only indicatory for us to evaluate how effectively is C. auris capable of developing resistance to azole substances. Therefore, each established strains’ MICs were compared strictly to their own parental strain. As drug resistance was developed strictly under laboratory conditions, these MIC values are not to be compared to those of freshly isolated clinical isolates.

Compared to the evolved strains the 0387 clinical isolate replicates at a slightly attenuated level. During the microevolution process the concentration of the antifungal drug was increased gradually starting from a sublethal dose to reach the final concentration. Following to this, evolved strains were serially inoculated in unsupplemented media before the antifungal testing, to make sure that the cells developed stable resistance and not just tolerance to the corresponding antifungal. Since we couldn’t see excessive differences in the antifungal susceptibility between the 24h (prominent difference) and 48h (plateau phase already reached) timepoints, we did not think that the difference of the growth capability hindered the MIC determination. Further, we have replicates in our experiments that behave similarly.

4) It is not clear what was the objective of the authors with the experiments shown in FIg.1. What was the rationale for this? Wouldn't this be better in supplementary material? By the way, growth curves should be log and the differences between the strains are very slight (with the exception of the growth deffect of strain 0387) which is more visible.

Thank you for the suggestion. As far as we know growth curves can be represented based on both absorbance and in logarithmic form. As no differences were found in the case of the 0381 strains, we moved that part of the Fig. 1 into the supplementary material as suggested. Since growth curve of the 0387 strains indicate the failure to adapt cell cycle to nutrition supply, which linked to the BCY1 mutation, we decided to keep this result in the main text. A short explanation about the reason of this experiment is now also included. As mentioned above, different C. auris clinical isolates have different characteristics, thus may differ in their growth abilities. So is the case for 0387 and 0381, both of which effectively grow in complex YPD medium, however one (0387) has a lower replicating rate, than the other (0381).

5) Cell wall perturbations and membrane perturbations were performed using an experimental setting different from the one used to evolve the strains. This may influence the response of the cells in a manner that is unrelated with the azole-resistance phenotype

Cell wall perturbations and membrane perturbations were performed using a specific, yet widely accepted and frequently applied method. The microevolved strains were generated through the microevolution process - in the presence of an increasing amount of drug concentration under complex YPD growing conditions.

To see if the acquired drug resistance caused any alteration in the cell wall and cell membrane of the strains, we next examined each strains’ cell wall and membranes composition: in complex YPD media (flow cytometry, fluorescent microscopy) and in cell wall and membrane perturbant supplemented complex YPD media (abiotic stress tolerance). As the stable drug resistant phenotype was already established, the corresponding drugs were not added to the complex YPD media during the subsequent analyses.

The routinely applied spotting assay was used to determinate how the acquisition of resistance changed the abiotic stress tolerance of the generated stains. As indicated in the manuscript unsupplemented YPD plates were used as a control in both studied temperatures to see possible growth differences between the clinical isolates, served as a base for the microevolution experiments (0381, 0387) and the generated resistant (FLUevo, POSevo, VORevo) strains. This provided us insights into, how both susceptible and resistant strains behaved in this environment without any perturbation. As we determined the subjective growth scores in supplemented plates, we first considered how strains were able to grow on the corresponding YPD control plates before we compared them to the originating susceptible clinical isolate. So, the experimental setting was indeed different, but the clinical isolates were exposed to the same conditions before we made the comparations. Interestingly on solid YPD plates we saw the same growth patterns as in liquid media, as there were no detectable differences in the 0381 set, but 0387 evolved strains had a growth advantage compared to the clinical isolate. By editing the text, we tried to clarify that in every case, evolved strains were compared to the clinical isolate under the same conditions.   

6) In line 508 authors aim to make the point that the abundance of byproducts suggests there is an alternative pathway to Erg11 when the cells are challenged with azoles. But...don't we know that this is the case and the reason why cells accumulate the methylated sterols? What is the point that the authors are trying to make here? This figure (which has very intersting results) is still very complex, maybe if the authors present the ratio btween what happns to sterols in the same strain after exposure to FLC and before this would become easier to follow. The identification of the new sterol seems very intersting, has this been described in other species?

Thank you, we changed the title according to the suggestion. We apologize that the corresponding figure was hard to follow. During the preparation of the manuscript, we tried several versions and this layout seemed the most clear. The current layout follows the logic of the text as azole treated (Figure 3A-B) and untreated strains (Figure 3C-D) are divided into different panels. Even so, we tried to make the figure more concise by introducing some color coding to indicate the same strains. To our knowledge portensterol was only found in Tricholoma species in the past, while stigmasterol was found in Pneumocystis carinii. It has been proposed in the past by Rybak et al. (2020, 2022) that C. auris cells can accumulate atypical sterols, such as cholesta-type products.  

7) while in the abstract argues about the multiplicity of mechanisms governing acquisition of azole resistance, the title of the section describing the genomics section argues the opposite. In fact, since the genes affected are not the same why the authors make this point for a single mechanisms of resistance? 

Thank you for this remark. We changed the title to be more representative of this part of the results.

8) Informatic predictions on how modifications affect proteins are much flawed and, in my view, should be disregarded specially in transcription factors (and the authors make this point in the discussion). Have these modifications in Tac1B been described elsewhere? in its homologues in other yeasts? To clearly associate these modifications with changes in protein activity, it is essential to undertake the necessary complementary functional assays using the wt Tac1B and the mutated one. Without that, it appears only as a speculation to say that it might be affected. What are the evidences for that?

Thank you for the observation. We removed the parts of the manuscript where predictions about transcription factors were made.

As suggested we also integrated further information from TAC1B in both the introduction and discussion parts of the manuscript. The role of this transcription factor was already confirmed in the azole resistance of C. auris by both knocking out the gene and by introducing the most common SNP (A640V) into previously azole susceptible isolates. Confirming the role and the effect of the amino acid substitutions we identified (N690S, S19I) in the generated strains is in our future plans, as the data in this work is already extensive.    

9) A deletion of 200 bp in 004089 gene was inside the coding sequence? If yes, this surely has an impact in protein activity! THis is wrongly presentd and must be explained so that we understand what happens to this protein, is it inactive? or active? Specially important considering the role of Rox in ergosterol metabolism in other yeasts.

Thank you for this remark. We modified the text to make the nature of the mutation more comprehensible. The plausible role of this mutation was also revised and edited in the manuscript. The nature of this mutation (200bp deletion) indicates inactive gene product, but this conclusion needs conformation. To this day function of this gene is not entirely clear in. In C. albicans the ortholog of  B9J08_004089 is RFG1. This gene mainly participates in morphogenesis, while its ortholog in S. cerevisiae, ROX1 regulates filamentous growth to some extent, but mainly responsible for the repression of hypoxia related genes, including genes that participate in ergosterol biosynthesis. There are some evidence that this gene can also have a role in hypoxia associated responses, as in C. glabrata the deletion of ROX1 can compensate the UPC2 deletion by regulating ERG genes. Because of this complexity, study of this gene in C. auris could be an fascinating future project.    

10)What it means an X substitution? Is the protein truncated? if yes, clarify this aspect please. what percentage of the protein remains ?

We indeed used an atypical annotation to indicate nonsense mutations. To solve this issue we modified corresponding parts of the manuscript. The marking of nonsense mutation was changed to a more conventional one, and further explanation about the truncated product were also included.

11) BCY1 is not a transcription factor...

Thank you for this remark. We corrected the mistake.

12) The discussion section is so long...it is very very difficult to understand what the authors want to convey as it seems that all things have to be connected. Please try to reduce this and aim for a message that we can take out this very interesting set of results. 

Thank you for this remark. We tried to be inclusive in our assessments of the results and this led to the prior length of the discussion To solve this issue, we tried to reduce this section by removing certain parts. The discussion was also divided into subsections and all subsections were summarized by a title. Also, in accordance with the author guide we included an additional chapter ‘Conclusions’ to further summarize the main points of the manuscript. We hope that these changes can make the text easier to follow and interpret.

Reviewer 2 Report

Comments and Suggestions for Authors

General Impression

The authors summarize an impressive body of work on the biology and pathogenicity of newly evolved azole-strains of Candida auris. As anyone working in Candida species knows, the organisms have remarkable genomic plasticity and phenotypic variation. The authors recognize the biological variation and pleiotropy of Candida genetic adaptations and present a suitable analysis of the observed phenomenon. The manuscript provides valuable data on the development of Candida auris drug resistance, but due to the complexity of the phenomenon cannot provide a comprehensive explanation of the observations. The experiments are well-planned, thoroughly executed and appropriately analyzed. The discussion reflects the ambiguity and preliminary nature of the findings and only occasionally veers into speculation. In this, the paper provides relevant data for the field and deserves publication.

Specific points

1.      Author byline – it appears that the name of the last author has been omitted as the sentence ends with “and”.

2.      DNA extraction , line 312. The listed centrifugal force (16.2 g) is way too low to pellet DNA precipitate. Could this be a misprint?

3.      Fungal burden of infected mice, Figure 10. There is a lot of variation in the colonization numbers for the 0387 strain. It would be helpful to information on total fungal burden as it could explain the variation – do the fungi preferentially colonize the liver of one animal and the liver of another?

Comments on the Quality of English Language

Generally well written, problems can be solved by copy editor.

Author Response

We thank the reviewers and the editor for their time and effort to evaluate our manuscript carefully. We appreciate all the comments and concerns raised during the review and hope our answers effectively address each point. 

  1. Author byline – it appears that the name of the last author has been omitted as the sentence ends with “and”.

Thank you for pointing this out. The additional ‘and’ was accidentally included in the editing process. 

  1. DNA extraction , line 312. The listed centrifugal force (16.2 g) is way too low to pellet DNA precipitate. Could this be a misprint?

We apologize the manuscript indeed contained the wrong parameters. We corrected these mistakes in the text.

  1. Fungal burden of infected mice, Figure 10. There is a lot of variation in the colonization numbers for the 0387 strain. It would be helpful to information on total fungal burden as it could explain the variation – do the fungi preferentially colonize the liver of one animal and the liver of another?

During our in vivo experiments, a few infected animals we had significantly elevated CFU levels in certain organs. Based on the literature these two organs (spleen and liver) possess a fairly similar immune response upon fungal infection. Therefore, we think that that if the level of the fungal colonization reaches a certain threshold, it can bypass the organ specific immune response and the infection become uncontrolled. We clarified this information to support this hypothesis.     

Reviewer 3 Report

Comments and Suggestions for Authors

- 2.14 No ethical approval statement is included, this is the minimum that is required and what body approved these experiments.

- Figure 2A and B: Slightly better growth and better growth, medium defect and strong defect is not a quantifiable term. I would prefer to see spot tests or the raw data from growth.

- 3.7 Cell wall composition is not directly assessed but binding of compounds to cell wall components. I prefer different language here.

- Figure 4: unpaired t-tests should not be used but ANOVA with multiple comparison. Else you will find false positives.

- 3.89 I would be amazed if the evolved strain only had one SNP in the entire genome. How many other SNPs found, how many are in coding regions etc. This information is required.

- A major drawback of this work is that mutations found in the evolved strains are not reconstructed via genetic editing in the parental to confirm the phenotypes found. So far this is all correlative data.

- Another major drawback is that no replication of microevolution is performed. Usually microevolution is an experiment of chance. Multiple replicates of the same strain under the same condition could lead to finding other mutations. Concluding that these are strain specific mutations is a step too far and can't be concluded from 1 replicate.

- For graphs that use Mann-Whitney tests median should be shown. It is not mentioned if the line shows average or median.

Author Response

- 2.14 No ethical approval statement is included, this is the minimum that is required and what body approved these experiments.

Thank you for the remark. In line with the publisher’s instructions the ethical statement is included under the ‘Institutional Review Board Statement’ subsection of the manuscript. We included the local (1998, XXVIII; 40/2013) and the European (2010/63/EU) animal ethics guidelines that we followed during the in vivo experiments, as well as the permission number (XX./511/2022) provided by the Animal Experimentation and Ethics Committee of the Biological Research Centre of the Hungarian Academy of Sciences and the Hungarian National Animal Experimentation and Ethics Board and the University of Szeged.

- Figure 2A and B: Slightly better growth and better growth, medium defect and strong defect is not a quantifiable term. I would prefer to see spot tests or the raw data from growth.

Thank you for the suggestion. The representative scoring system was already used in our previous publications although, we agree that it is not quantifiable and can be regarded as rather subjective. Nevertheless, for each condition we compared the growth capability of the evolved strain to the clinical isolate, while also considering the possible differences of growth capability based on the YPD control plates (without any supplement). To make the scoring system easier to comprehend we included representative images of the spot assays in Figure 2 C, D, E, F panel. Because of the subjective nature of the scoring system, mind differences were not considered as significant during the further evaluation of the results, therefore we only discussed the clear differences where growth loss or gain was more evident. Images of the growth capability of the strains in every studied condition where a difference was found were also submitted as Appendix A. In the resubmission we also included images of the spotting assays where no difference was observed compared to the clinical isolate.

- 3.7 Cell wall composition is not directly assessed but binding of compounds to cell wall components. I prefer different language here.

Thank you for this suggestion. We modified the text accordingly.

- Figure 4: unpaired t-tests should not be used but ANOVA with multiple comparison. Else you will find false positives.

Thank you for the suggestion. We changed the statistical test and adjusted the figure and the text accordingly.

- 3.89 I would be amazed if the evolved strain only had one SNP in the entire genome. How many other SNPs found, how many are in coding regions etc. This information is required.

Thank you for this remark. To limit the length of the manuscript we only included the nonsynonymous SNVs (single nucleotid variations) that were detected during the genome analysis. To clarify this, we modified the text accordingly. When synonymous mutations and nucleotid modifications in non-coding regions are also considered, the number of SNPs were definitely higher. The 0381 POSevo also contained synonymous mutations in B9J08_001296 and B9J08_001296. In B9J08_001296 multiple nucleotid exchange was noted although none of them caused amino acid modification. Similarly, this gene was also modified in the VORevo strain. In 0387 azole evolved strains we could not identify synonymous SNVs in coding regions. Besides this, several SNPs were also identified in non-coding regions in each evolved strain.

- A major drawback of this work is that mutations found in the evolved strains are not reconstructed via genetic editing in the parental to confirm the phenotypes found. So far this is all correlative data.

Thank you, we agree that the work has some drawbacks. One of them is indeed the lack of confirmation of the identified mutations. We are currently working on generating the mutant strains to validate the sequencing results. As the length of the manuscript is already excessive and concentrates more on the effect of the acquired resistance, we are planning on publishing the phenotyping of the mutant strains as a follow up paper. Although we addressed this issue in the manuscript, we edited the text to make this weakness more apparent.  

- Another major drawback is that no replication of microevolution is performed. Usually microevolution is an experiment of chance. Multiple replicates of the same strain under the same condition could lead to finding other mutations. Concluding that these are strain specific mutations is a step too far and can't be concluded from 1 replicate.

Thank you for the comment. We modified the text to further highlight that the findings are showing a potential way that the acquired resistance can occur, and they are nowhere near definitive.   

- For graphs that use Mann-Whitney tests median should be shown. It is not mentioned if the line shows average or median.

Thank you for the observation. We adjusted the figures accordingly and also indicated the nature of the error bars in the figure legends.

Reviewer 4 Report

Comments and Suggestions for Authors

 Comments:

1. In table 1 and table 2, the authors detected the MIC of AMB, FLU, POS, VOR, AND, CAS, but why there is MIC in the last column? Dose the authors want to express “MICA”?

2. The resolution of figure 1 and figure 3 is relatively low. Please provide the figures with higher resolution.

3. It is suggested to used three-line tables in the text.

4. English writing is suggested to be polished. 

Comments on the Quality of English Language

English writing needs to be improved.

Author Response

  1. In table 1 and table 2, the authors detected the MIC of AMB, FLU, POS, VOR, AND, CAS, but why there is MIC in the last column? Dose the authors want to express “MICA”?

Thank you for the remark. It was indeed a mistake and the ‘MIC’ in the table was supposed to be MICA. We corrected the table accordingly.

  1. The resolution of figure 1 and figure 3 is relatively low. Please provide the figures with higher resolution.

Thank you for the observation. In accordance with the submission guide we uploaded figures in an .eps format, while in the manuscript template we embedded the figures in .jpg format. We are not sure what causes this issue, but we also exported both figures in .tiff format (300dpi; >1000 pixels) and included it in submission.

  1. It is suggested to used three-line tables in the text.

Thank you for the remark. We modified the table according to the suggestion. Since the tables containing the sequencing data are more complex, we used some additional lines there. We hope that it will not complicate the editing process.

  1. English writing is suggested to be polished.

Thank you for the suggestion. The manuscript has been corrected by a native speaker.

Round 2

Reviewer 1 Report

Comments and Suggestions for Authors

Although the responses are detailed (and parts of the text of the MS have also been improved) they merely try to make some rebuttal of the arguments while not providing experimental data that indeed may surpass some of the issues raised (this is the case of the results obtained with cell wall perturbing agents, merely to argue that the experiments were made in a way that people accept it, doesn’t really address the problem that the environment can modulate the response). The comments, in my view, should have originated some experimnts to make valid some points raised. I also can’t quite understand what is different in this work concerning evolution since the experiments seem to have been conducted and using strains used before; and the authors couldn’t really detail on why they work is novel. The gnomic analysis ends up to be quite confusing in the middle of everything else. On and on, i  would maintain my recommendation for reaction of the MS at this stage and an overall modification of it that could turn it more clear and emphatic on the aspects that are, indeed, novel. In this version these aspects aren’t clear.

Comments on the Quality of English Language

/

Author Response

Thank you for your continued engagement with our manuscript. We further edited the text according to the suggestions.

Regarding the novelty of our work, we have clarified in the revised manuscript how our approach, despite using known strains, contributes new insights into evolutionary processes.